

# Genetic structure and relatedness of brown trout (*Salmo trutta*) populations in the drainage basin of the Ölfusá river, South-Western Iceland

Marcos Lagunas[1], Arnar Pálsson[1], Benóný Jónsson[2], Magnús Jóhannsson[2], Zophonías O. Jónsson[1] and Sigurður S. Snorrason[1]

[1] Faculty of Life and Environmental Sciences, University of Iceland, Reykjavík, Iceland
[2] Marine and Freshwater Research Institute, Selfoss, Iceland

Corresponding author
Marcos Lagunas, mgl3@hi.is

## ABSTRACT

**Background:** Lake Þingvallavatn in Iceland, a part of the river Ölfusá drainage basin, was presumably populated by brown trout soon after it formed at the end of the last Ice Age. The genetic relatedness of the brown trout in Þingvallavatn to other populations in the Ölfusá drainage basin is unknown. After the building of a dam at the outlet of the lake in 1959 brown trout catches declined, though numbers have now increased. The aim of this study was to assess effects of geographic isolation and potential downstream gene flow on the genetic structure and diversity in brown trout sampled in several locations in the western side of the watershed of River Ölfusá. We hypothesized that brown trout in Lake Þingvallavatn constituted several local spawning populations connected by occasional gene flow before the damming of the lake. We also estimated the effective population size ($N_E$) of some of these populations and tested for signs of a recent population bottleneck in Lake Þingvallavatn.

**Methods:** We sampled brown trout inhabiting four lakes and 12 rivers within and near the watershed of River Ölfusá by means of electro- and net- fishing. After stringent data filtering, 2,597 polymorphic loci obtained from ddRADseq data from 317 individuals were ascertained as putative neutral markers.

**Results:** Overall, the genetic relatedness of brown trout in the Ölfusá watershed reflected the connectivity and topography of the waterways. Ancestry proportion analyses and a phylogenetic tree revealed seven distinct clusters, some of which corresponded to small populations with reduced genetic diversity. There was no evidence of downstream gene flow from Lake Þingvallavatn, although gene flow was observed from much smaller mountain populations. Most locations showed low $N_E$ values (*i.e.*, ~14.6 on average) while the putative anadromous trout from River Sog and the spawning population from River Öxará, that flows into Lake Þingvallavatn, showed notably higher $N_E$ values (*i.e.*, 71.2 and 56.5, respectively). No signals of recent population bottlenecks were detected in the brown trout of Lake Þingvallavatn.

**Discussion:** This is the first time that the genetic structure and diversity of brown trout in the watershed of River Ölfusá have been assessed. Our results point towards the presence of a metapopulation in the watershed of Lake Þingvallavatn, which has

been influenced by restoration efforts and is now dominated by a genetic component originated in River Öxará. Many of the locations studied represent different populations. Those that are isolated in headwater streams and lakes are genetically distinct presenting low genetic diversity, yet they can be important in increasing the genetic variation in downstream populations. These populations should be considered for conservation and direct management.

## INTRODUCTION

Changes in key climatic factors are predicted to be more pronounced in higher latitudes (*Hoegh-Guldberg et al., 2018*), and organisms that cannot respond by moving along gradients of latitude and altitude will be more affected (*Chen et al., 2011*). Potentially this applies to many populations of brown trout (*Salmo trutta*) in subarctic freshwater systems, such as those found in Iceland. This species most likely colonized Iceland soon after the end of the last Ice Age 10–14,000 years ago. During this period of ice cap retreat, the newly formed watersheds were further shaped by geological forces, tectonic movements, active volcanism, and erosion. Apart from diverting some waterways, these processes often caused formation of new lakes and impassable waterfalls thus preventing gene-flow from downstream to upstream populations.

Presently, brown trout is found in both lakes and streams in all parts of Iceland. Several populations, especially those inhabiting headwaters, have become isolated above waterfalls. Many systems with access to the sea host populations that no longer migrate to the ocean, *i.e.*, they have become permanent freshwater residents, while in other cases trout still retain the anadromous life cycle.

Lake Þingvallavatn (Fig. 1) has long been known for its brown trout (*Sturlaugsson & Malmquist, 2011*). This iconic population produced individuals of extreme size and was much appreciated by wealthy European anglers already in the early 19<sup>th</sup> century. They travelled to Iceland to experience angling of large trout at the outlet of the lake and in the outflowing river, Efra Sog, that runs in a 1.2 km long canyon into the smaller lake Úlfljótsvatn (*Malmquist, 2011*).

In 1959 a small dam was built on the outlet of Þingvallavatn as a part of the Steingrímsstöð hydro-electric power plant, taking advantage of the 22 m difference in elevation between the lakes and the stable discharge rate. This diverted the outflow through a tunnel and left the Efra Sog canyon dry apart from a small spring entering mid-ways between the lakes. The dam and associated disturbance presumably impacted the brown trout population of both lakes in several ways. The drying-up of River Efra Sog ruined an extremely productive benthic community with high density of black fly larvae (*Simulium vittatum*) (*Jóhannsson, Jónsson & Jónsson, 2005*). This habitat was likely optimal feeding grounds for juvenile trout (*Gíslason, Steingrímsson & Gudbergsson, 2002*). It is also

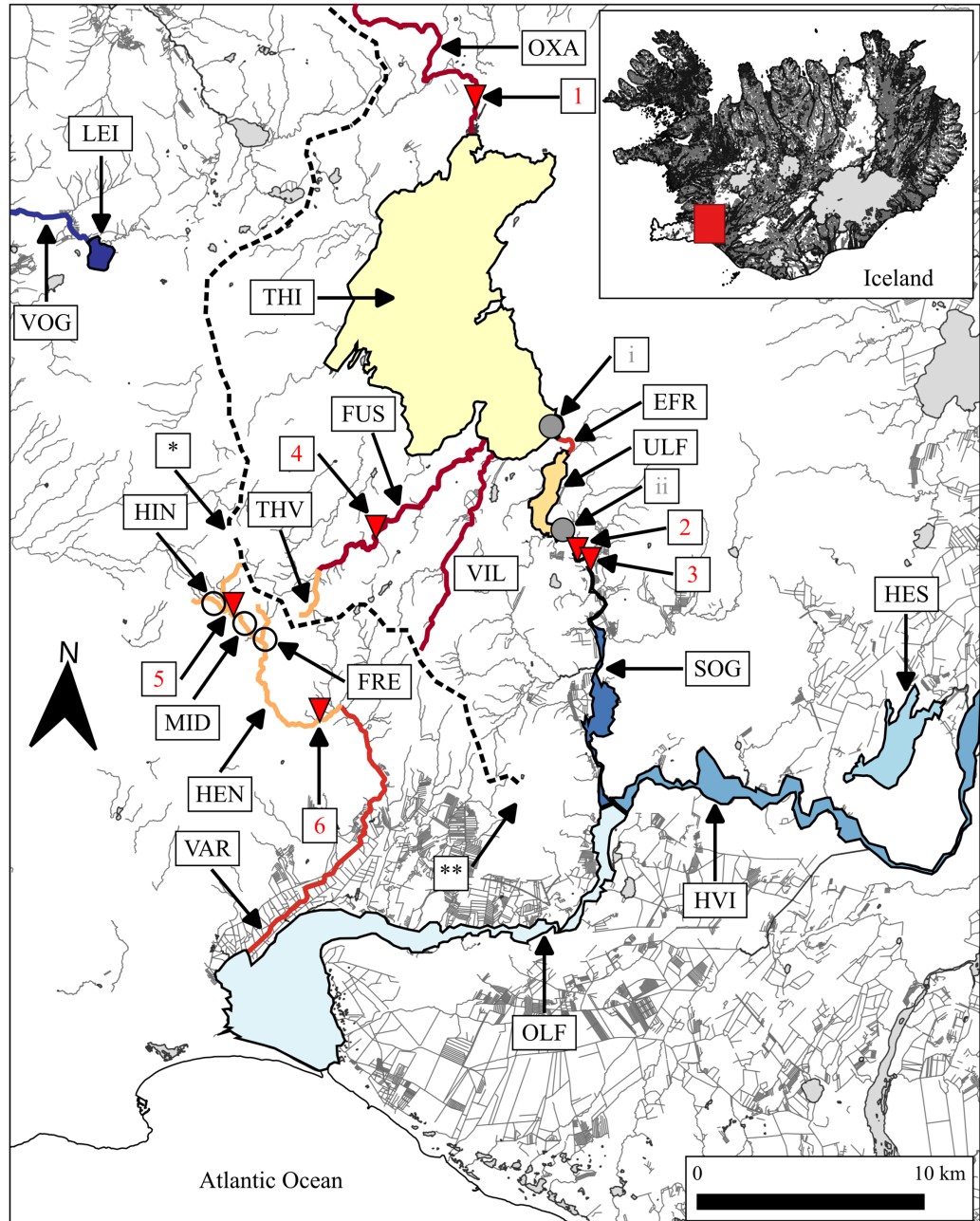

**Figure 1 Map showing rivers and lakes where the sampling of brown trout was conducted.** EFR, River Efra Sog; River Hengladalsá at Fremstidalur valley; FUS, River Ölfusvatnsá; HEN, River Hengladalsá; HES, Lake Hestvatn (connects to River Hvítá); River Hengladalsá at Innstidalur valley; HVI, River Hvítá; LEI, Lake Leirvogsvatn; River Hengladalsá at Miðdalur valley; OLF, River Ölfusá; OXA, River Öxará at Þingvellir National Park; SOG, River Sog; THI, Lake Þingvallavatn; THV, River Þverá; ULF, Lake Úlfl-jótsvatn; VAR, River Varmá; VIL, River Villingavatnsá; VOG, River Leirvogsá (drains to the west). The inverted red triangles show natural waterfalls. 1, Öxarárfoss; 2, Írafoss; 3, Kistufoss; 4, unnamed waterfall separating the lower Ölfusvatnsá from the upper Ölfusvatnsá and Þverá rivers; 5, unnamed waterfall in Hengladalsá situated between the valleys Innstidalur and Miðdalur; 6, unnamed waterfall in Hengladalsá at Kambabrún just before it joins Varmá (downstream). The grey circles indicate the location of dams (i, at the outflow of Þingvallavatn; ii, at the outflow of Úlfljótsvatn). *, Skeggi, the highest

**Figure 1 (continued)**
peak in the Hengill volcano at 799 m above sea level; **, Inghóll, the highest peak of the Ingólfsfjall mountain at 551 m above sea level. The dashed line represents the approximate position of the drainage divide. This map was generated with the program QGIS (*QGIS Development Team, 2023*) using data from the National Land Survey of Iceland (*LMI, 2020*). In Icelandic, the suffix-á indicates a river or stream, while the suffix-vatn indicates a lake. From here on there will not be a clarification following or preceding names of water bodies in the text.               

commonly accepted that the construction of the dam and the tunnel destroyed a major spawning site of trout at the outlet (*Sturlaugsson & Malmquist, 2011*).

The population structure of the brown trout in the watershed of Þingvallavatn before 1959 can only be inferred from records from local fishermen and anglers (*Sturlaugsson & Malmquist, 2011*). According to those sources, there were two main spawning grounds in Þingvallavatn, in Öxará, a river that enters the lake on the northern shore, and at the outflow at Efra Sog (Fig. 1). Brown trout also spawned in the lower reaches of Ölfusvatnsá and Villingavatnsá, two smaller rivers that enter the lake on the southern coast (*Skarphéðinsson, 1996*), and considering the strong homing tendency of salmonids, they probably constituted separate sub-populations. Downward gene flow from an isolated mountain population (*i.e.*, in the Hengill volcano) in the Þverá/upper Ölfusvatnsá rivers (*i.e.*, above an impassable waterfall in Ölfusvatnsá) may have influenced the genetic composition of the brown trout of Þingvallavatn and Úlfljótsvatn, especially the spawning populations in the lower Ölfusvatnsá and the nearby Villingavatnsá. Hence, if the waterways of Þingvallavatn and Úlfljótsvatn had not been altered we would expect the genetic structure to reflect a metapopulation where several local spawning populations were connected by occasional gene flow (*Hanski, 1998*).

Although the major spawning grounds at the outlet were destroyed by the damming, spawning of trout would not have been directly affected in other locations, *e.g.*, in Öxará, Ölfusvatnsá, and Villingavatnsá. Yet catches of trout in Þingvallavatn declined sharply in the 1960s and 1970s, especially in the southern and eastern parts of the lake (*Skarphéðinsson, 1996*; *Sturlaugsson & Malmquist, 2011*). In two lake-wide surveys of fish in 1981 and 1984 only a handful of trout were caught (*i.e.*, 15 of 2,206 fish or 0.68% of the catch in 1981 and 5 of 3,715 fish or 0.13% in 1984) (*Malmquist, Snorrason & Skúlason, 1985*; *Sandlund et al., 1987*). A decade later the trout in Þingvallavatn still showed no signs of recovery (*Guðbergsson & Guðjónsson, 1993*; *Guðbergsson, Guðjónsson & Jóhannsson, 1994*). These data strongly indicated general, prolonged recruitment failures. Catches of trout in Úlfljótsvatn also diminished to low numbers (*Jóhannsson & Jónsson, 2000a*) after 1959, suggesting severe population bottlenecks in both lakes.

Several attempts were made to re-establish the trout in Þingvallavatn. In 1993, 10,000 one year old trout derived from Öxará spawners (three females and four males) were released at many locations in Þingvallavatn (*Guðbergsson, Guðjónsson & Jóhannsson, 1994*). Between 1999 and 2004, 6,550 and 3,600 fertilized eggs were planted in Þingvallavatn and Ölfusvatnsá, respectively. Moreover, 119,000 fingerlings (0+) and 3,900 one year old juveniles were released in Þingvallavatn near the outflow. All parr and eggs

originated from spawners caught in Öxará (*Guðbergsson, Guðjónsson & Jóhannsson, 1994*; *Jóhannsson & Jónsson, 2000b*). These efforts to recuperate the brown trout stock seem to have been successful as the number of spawning fish in Ölfusvatnsá increased significantly in the period 2003–2009 (*Jóhannsson & Jónsson, 2015*). The spawning population of Öxará also showed definite signs of recovery (*Jóhannsson, Jónsson & Jónsson, 2005*; *Jóhannsson & Jónsson, 2015*; *Jóhannsson & Jónsson, 2016a*, *2016b*). Catches of brown trout in Þingvallavatn have increased steadily in the last two decades and a survey of relative fish densities in 2019 showed the proportion of brown trout amounting to 3.1% at the eastern shore, 5.6% at the northern shore, and 67% at the southern shore near the outflow of Ölfusvatnsá (Q. J. B. Horta-Lacueva, F. Finn, F. Ingimarsson, H. R. Ingvarson, H. Xiao, K. H. Kapralova, L. Ponsioen, M. Lagunas, M. De La Camara, N. Eskafi, S. M. Stefánsson, S. S. Snorrason, 2022, unpublished data).

In 1993, Landsvirkjun, the company that operates the Steingrímsstöð power plant, changed its operation of the dam by allowing 3–4 m$^3$/s to flow on average through a bottom sluice to the natural outflow, *i.e.*, the Efra Sog canyon (*Jónsson & Jóhannsson, 2012*). This restored a limited proportion of the plant and invertebrate communities of the Efra Sog canyon and allowed fish from Úlfljótsvatn to utilize this habitat. Currently the company is evaluating plans to build a fish ladder beside the dam thus allowing brown trout to migrate between the lakes. Little is known about migration patterns of trout between Þingvallavatn and Úlfljótsvatn prior to the dam construction. Considering the topography of the Efra Sog canyon there is no reason to believe that the connection between the lakes presented a strong physical barrier for downward or upward movements of fish. Hence, we do not expect to find significant genetic differences between trout from Úlfljótsvatn and Þingvallavatn.

It should be noted that brown trout in these lakes were already isolated from populations downstream by two waterfalls that likely formed shortly after the end of the last Ice Age, *i.e.*, Írafoss and Kistufoss, just below the outflow of Úlfljótsvatn, 60 and 50 m above sea level, respectively (Fig. 1).

Þingvallavatn and Úlfljótsvatn are a part of a system of waterways, most of which harbour brown trout. Several streams and rivers originate in the slopes of the Hengill volcano, a central volcano located to the south-west of Þingvallavatn (Fig. 1). On the eastern slopes, streams such as Þverá drain into Þingvallavatn *via* River Ölfusvatnsá while on the southern slopes they drain into River Hengladalsá, which, after dropping through a series of waterfalls, joins River Varmá. These mountain streams presently sustain small brown trout that are isolated above impassable waterfalls. Many of those streams are influenced by geothermal warming, with temperatures ranging from 4 °C to 24 °C (*O'Gorman et al., 2016*; *Ólafsson, 2019*). The origin and genetic relationships of these populations are unknown. Presumably, the trout in Þverá originated from the population that colonized Þingvallavatn. Humans may also have played a role in their establishment in Hengladalsá since there are records of brown trout being released in Varmá during the 1960s (*Guðjónsson, Guðmundsson & Jónsson, 1983*).

Brown trout in the lowland watersheds of Sog, Hvítá, and Varmá have potential access to the ocean *via* Ölfusá (Fig. 1) and are referred to (from here on) as putative anadromous

populations, as suggested by *Jóhannsson et al. (2020)*. The extent to which these lowland populations might be influenced by gene flow from upstream brown trout in Þingvallavatn and Úlfljótsvatn or the mountain streams of the Hengill volcano is unknown.

Here we use ddRADseq data to assess the genetic structure and relatedness of brown trout in several locations in the western side of the drainage basin of the River Ölfusá.

We also assess the effects of geographic isolation on the genetic diversity of trout sampled at these locations and estimate their effective population size. Furthermore, we evaluate potential downstream gene flow from the isolated populations of streams in the slopes of the Hengill volcano as well as from the brown trout of Þingvallavatn and Úlfljótsvatn into lowland populations. Finally, we evaluate whether the brown trout of Þingvallavatn experienced a population bottleneck after the damming of the lake.

We hypothesize that the brown trout inhabiting Lakes Þingvallavatn and Úlfljótsvatn constituted a metapopulation before the damming of Þingvallavatn, with local sub-populations spawning in the tributaries (*i.e.*, Öxará, Ölfusvatnsá, Villingavatnsá) and at the outlet (*i.e.*, Efra Sog). Our results will shed light on the genetic relatedness and connectivity of these brown trout populations and assist authorities in their efforts of conservation and management.

## MATERIALS AND METHODS

### Sample collection

We sampled 555 brown trout from a total of four lakes and 12 rivers in and around the drainage basin of River Ölfusá, including Lakes Þingvallavatn and Úlfljótsvatn (Fig. 1 and Table 1). Samples of large fish in either lakes or rivers were obtained using nylon gill-nets (25 m × 1.5 m Lundgren series, mesh sizes ranging from 25.0 to 50.0 mm). When used in lakes, the nets were deployed from a boat close to the lake shore where the depth is more than 1.5 m. The nets were set perpendicularly to the coastline and left overnight. Fish were collected after 12 h of fishing, killed, and either processed *in situ* or taken to the laboratory on ice.

Spawners were sampled in two rivers, Öxará and the lower Ölfusvatnsá. In the latter, nets of mesh size between 60–65 mm were stretched between shores and dragged upstream until up to five individuals were caught. In Öxará, spawners were caught in traps. The fish were sedated with 2-phenoxyethanol (*Pounder et al., 2018*), and fin clips taken for DNA analysis. The fish were released after having recovered for 30 min in a trap net. Samples of small fish in rivers were taken by electro-fishing. Gear from KC Denmark (Silkeborg, Denmark) was used in the sampling of rivers and streams. It consists of two electrodes (one attached to a 2 m pole) powered by a gasoline engine that produces a current at 200 V AC. The output is run through an AC/DC converter and the resulting current can be applied continuously or pulsed. The stunned fish were collected with a dip net and processed at the sampling location. The fish were immediately sedated upon collection with 2-phenoxyethanol so that a small fin clip could be collected for DNA analysis. The fish were returned to their habitat after they had recovered in a bucket of water with no sedative.

**Table 1 Sampling (approach and locations) of brown trout, with focus on the watershed of River Ölfusá.**

| Name | Code | Habitat | Coordinates | Method[1] | Ds*[2] | n[3] | Year[4] |
|---|---|---|---|---|---|---|---|
| Efra Sog | EFR | River | 64°07.9555N; 21°01.3643W | E | L | 15 | 2015, 2017 |
| Hengladalsá (Innstidalur) | HIN | River | 64°03.5306N; 21°19.1977W | E | L* | 6 | 2015 |
| Hengladalsá (Fremstidalur) | FRE | River | 64°02.7940N; 21°17.2070W | E | L | 16 | 2016 |
| Hengladalsá (Miðdalur) | MID | River | 64°03.1437N; 21°18.2141W | E | L | 70[5] | 2015, 2016 |
| Hestvatn | HES | Lake | 64°02.5780N; 20°41.6820W | N | L | 38 | 2015, 2017 |
| Hvítá | HVI | River | 64°00.0448N; 20°47.6172W | A | U | 7 | 2015, 2017 |
| Leirvogsvatn (local reference) | LEI | Lake | 64°11.8965N; 21°27.7426W | N | L* | 65 | 2015 |
| Sog | SOG | River | 64°00.2804N; 20°58.3753W | E | L | 24 | 2015 |
| Úlfljótsvatn | ULF | Lake | 64°07.3151N; 21°01.6773W | N | U | 32 | 2015 |
| Varmá | VAR | River | 63°59.2883N; 21°10.3639W | E | L | 18 | 2015 |
| Villingavatnsá | VIL | River | 64°07.2673N; 21°05.7015W | E | L | 11 | 2015, 2017 |
| Þingvallavatn | THI | Lake | 64°11.3941N; 21°05.5264W | N | U | 60[5] | 2015, 2016, 2017 |
| Þverá/upper Ölfusvatnsá | THV | River | 64°04.1278N; 21°14.5523W | E | L* | 36 | 2016 |
| Lower Ölfusvatnsá | FUS | River | 64°07.6252N; 21°06.5245W | E, N | L, S* | 44[5] | 2015, 2017 |
| Ölfusá | OLF | River | 63°56.5361N; 20°59.8801W | E | U | 18[5] | 2017 |
| Öxará | OXA | River | 64°15.8210N; 21°07.2659W | T | S* | 31 | 2015 |
| Loch Slapin (reference group) | SLP | Marine loch | 57°10.4914N; 06°01.3950W | N | U | 64 | 2017 |
| TOTAL | | | | | | 555 | |

**Note:**
[1] The collection was done with electro-fishing (E), net fishing (N), trapping (T), and angling (A).
[2] Presumed definition of sample (Ds*) i.e., S, spawners;, L, local juveniles and adults; U, individuals from unidentified population. An asterisk (*) was added to those locations where it is presumed that the individuals sampled represent actual spawning populations, either because spawning fish were sampled or because the size and characteristics of the habitat strongly suggest it, i.e., isolated, upstream, and very small.
[3] Number of samples sequenced per location.
[4] Sampling year.
[5] The sampling locations Hengladalsá (Miðdalur), Þingvallavatn, lower Ölfusvatnsá, and Ölfusá, represent a combination of several sampling stations in close proximity to each other within the same water body. A preliminary analysis showed that there were no genetic differences between stations and thus were combined into a single location.

It is not possible to consider individuals from each site as samples from a spawning population since we cannot ascertain that all fish originate from the river or lake where they were caught. However, in the cases where spawning fish were sampled on spawning grounds, or where the habitat is very small and isolated, we assumed that these samples represented proper spawning populations (Table 1).

Sigurður S. Snorrason, lead author on this study has a scientific salmonid fieldwork permit issued by the Directorate of Fisheries, #0460/2021-2.0 and #0042/2014-2.13. All fishing in Lake Þingvallavatn was done with permissions obtained from the farmers in Mjóanes and from the Þingvellir National Park Commission and directors of the park Ólafur Örn Haraldsson and Einar Á. E. Sæmundssen. Fishing in Hestvatn was done with the permit and help of Sigrún Reynisdóttir (landowner at Eyvík farm). Sampling in other rivers was done within the Freshwater Institute monitoring regime of salmonids in Varmá, Sog, Ölfusá, Efra Sog, Villingavatnsá, Ölfusvatnsá, and Leirvogsvatn. The Hengill volcano streams were surveyed with Gísli Már Gíslason (professor emeritus). Samples from Hvítá were provided by Árni Guðmundsson, farmer at Arnarbæli, caught as by-catch in salmon nets.

An anadromous population from Loch Slapin (Isle of Skye, Scotland) was chosen as a reference group in our analyses. The samples from this population were kindly provided by Dr. Colin Adams from the University of Glasgow.

## DNA isolation

The DNA extraction was performed using a NucleoMag Tissue purification kit (Macherey-Nagel, Düren, Germany). Fin clips or muscle tissue samples were submerged in a solution containing proteinase K and lysis buffer and digested overnight at 55 °C. Immediately prior to removing the supernatant, a secondary digestion with RNase was performed for 5 min at room temperature. Samples were centrifuged at $5,600 \times g$ for 5 min and the supernatant was retrieved. A suspension of magnetic beads was added, mixed with the sample by pipetting, and allowed to incubate for 5 min at room temperature. After a series of washes, the mixture was incubated in elution buffer (5 mM Tris/HCl, pH 8.5) for 10 min at 55 °C. The supernatant was separated using a magnetic plate and the concentration of DNA was measured *via* spectrophotometry (Nanodrop). The DNA quality was assessed *via* gel electrophoresis using 0.6% agarose gels run at 200 V for 45 min. Low quality samples were discarded to avoid preparation of libraries with highly sheared DNA, which would yield a low number of raw reads and possible PCR artifacts (*Graham et al., 2015*).

## Library preparation and sequencing

The methodology is based on *Peterson et al. (2012)*. For every ddRAD library, 96 DNA samples from at least eight different locations were equalized to the same concentration (*i.e.*, 30 or 50 µg µl$^{-1}$) and distributed in different rows of 96-well plates. Sample DNA was digested with *Bam*HI-HF (final concentration 0.16 U µl$^{-1}$) in CutSmart buffer (New England Biolabs) for 4 h at 37 °C (final reaction volume 30 µl). After adding NEB 3.1 buffer (New England Biolabs) to the first digestion mixture, a second digestion with *Ape*KI (final concentration of 0.04 U µl$^{-1}$) followed immediately for 4 h at 75 °C (final reaction volume 50 µl). Eight adaptors for *Bam*HI-HF cut sites were combined with 12 adaptors for *Ape*K cut sites to produce 96 unique combinations which were assigned to each sample (Table S1). Adaptors were mixed with samples and T4 ligase (final concentration of 40 U µl$^{-1}$) and incubated for 12 h at 21 °C. The enzyme was inactivated by incubating the mixture at 65 °C for 10 min.

All 96 samples were pooled together and purified using magnetic beads (Macherey-Nagel, Düren, Germany). Fragments of the appropriate size (*i.e.*, 360–440 bp) were isolated using a 2% agarose Pippin Prep gel cassette and an internal marker for enhanced accuracy (Sage Science, Beverly, MA, USA). A PCR reaction to amplify the resulting fragments was performed using OneTaq reaction mixtures (New England Biolabs) in the following conditions: 72 °C for 3 min, 98 °C for 30 s, 11 cycles of 10 s at 98 °C, 30 s at 65 °C, and 45 s at 72 °C, and a final extension time of 5 min at 72 °C. A further clean-up step was performed with magnetic beads (Macherey-Nagel, Düren, Germany) and the final product was eluted in 5 mM Tris/HCl buffer (pH 8.5) and visualized *via* electrophoresis in a 2% agarose gel, run for 45 min at 80 V. The final DNA concentration was measured using Qubit fluorometric quantification with the dsDNA BR Assay (Life Technologies, Carlsbad,

CA, USA). Library quality was assessed by sequencing on an Illumina MiSeq platform (2 × 150 bp paired-end reads) in four out of the total 13 libraries prepared. Data from MiSeq and HiSeq runs were later combined. Each library was sequenced on a single lane of an Illumina X Ten platform (2 × 150 bp paired-end reads) at the BGI Tech Solutions facility in Hong Kong.

## Data analysis

### Read processing and assembly of loci

After an initial quality check of the raw reads with the software *fastqc* (*Andrews et al., 2019*), we demultiplexed and trimmed them to 115 bp using the script *process_radtags* from the package *Stacks* v2.4 (*Catchen et al., 2011*). We followed a *de novo* approach running all programs manually within *Stacks*. This dataset was created before the publication of the *Salmo trutta* genome, part of the 25 Genomes Project (*Hansen et al., 2021*), available at the National Centre for Biotechnology Information website (RefSeq assembly accession: GCF_901001165.1). However, the results from SNPs obtained with a reference genome approach were very similar to the findings presented here (Fig. S1). We determined the optimal values for the parameters M (*i.e.*, number of mismatches between the two alleles of a heterozygote) and n (*i.e.*, number of mismatches between two alleles in a population, as defined by sampling location) before assembling putative loci, as suggested by *Rochette & Catchen (2017)*. After deciding that M = $n$ = 2, we ran the program *ustacks* to identify putative alleles using the parameter m = 3 (*i.e.*, number of identical reads to call an allele). Before proceeding to the building of a catalogue of consensus loci with the program *cstacks*, samples with an average coverage <20× were discarded. The catalogue was built using three samples from 26 different locations (26.64× average coverage). The programs *sstacks*, *tsv2bam* and *gstacks* were successively run to match all samples to the catalogue, incorporate paired-end reads, and align reads per sample calling variant sites in the populations, respectively. The program *populations* was run with a minimum threshold of 66% for a locus to be present in the individuals of each population, a minimum minor allele frequency of 5%, and a maximum observed heterozygosity of 60%. A description of the function as well as the values assigned to parameters of each program used at this stage can be seen in Table S2.

### Data filtering

The script *paralog-finder* v1.0 (*Ortiz, 2018*) was used to identify and discard paralogous sequences based on the methodology from *McKinney et al. (2017)*, which uses (i) the higher expected proportion of heterozygous individuals for duplicate loci and (ii) the deviation from the expected 1:1 allele ratio for singleton loci per heterozygous individual, to identify paralogous loci. *Populations* was run again using a blacklist of paralogous loci identified by *paralog-finder* and the flag—write-single-SNP to keep only the first SNP of each locus and avoid bias from treating physically linked SNPs as independent. We removed individuals with first-degree kinship using the software *vcftools* v0.1.16 (*Danecek et al., 2011*) with the flag—relatedness2 to prevent any influence from highly

related individuals (especially in locations sampled by electro-fishing). This tool calculates a relatedness statistic based on *Manichaikul et al. (2010)*, for pairs of individuals.

Next, *vcftools* v0.1.16 was used to remove loci with a minor allele count lower than three, a minor allele frequency lower than 5%, and genotypes with less than three counts. A threshold of 60% was used to remove individuals missing more than that proportion of data. Similarly, loci missing in more than 40% of the individuals from each sampling location were removed. To ascertain selectively neutral loci, we removed loci that were significantly out of Hardy-Weinberg equilibrium ($p < 0.05$) within any of the sampling locations using the scripts from *Puritz, Hollenbeck & Gold, 2014*. The values for parameters used in these data filtering steps, and the resulting number of loci, are summarized in Table S2, and the number of removed individuals at every filtering step is shown in Table S3.

### Population structure and genetic diversity analyses

Several packages of the program *R* (*R Core Team, 2022*) were used. Routines from the package *adegenet* v1.3 (*Jombart & Ahmed, 2011*) were used for principal component analysis (PCA) and to study genetic differentiation of populations (defined as sampling locations). The function *fst_WC84* from the package *assigner* (*Gosselin, 2020*) was used to calculate pairwise $F_{ST}$ values based on *Weir & Cockerham (1984)*. The $F_{ST}$ and confidence intervals were estimated with bootstrap of markers. The package *hierfstat* (*Goudet, 2004*) was used to calculate the observed and expected heterozygosities (*i.e.*, $H_O$ and $H_E$) and the number of alleles and loci present in every location. The function *pi* from the package *radiator* (*Gosselin et al., 2020*) was used to calculate the nucleotide diversity per location (*Nei & Li, 1979*).

The function *snmf* from the package *LEA* (*Frichot & François, 2015*) was used to estimate admixture coefficients using sparse non-negative matrix factorization. This function calculates an entropy criterion that can be used to estimate the number K of ancestral populations, as in general the smaller the value of the criterion the better algorithm estimation (*Frichot et al., 2014*). We ran it 2,500 times for every K value between 7–13 and chose the run and K value that minimized the entropy criterion. The function *floating.pie* from the package *plotrix* (*Lemon, 2006*) was used to plot the admixture proportions of ancestral populations for a chosen K value on a map of the study area.

The program *RAxML* v8.2.11 (*Stamatakis, 2006*) was used to estimate the phylogenetic relationship between locations by creating a maximum likelihood tree. The GTR-GAMMA model of substitution was used, and the best scoring tree was chosen using a frequency-based criterion and the rapid bootstrapping algorithm described in *Stamatakis, Hoover & Rougemont (2008)*.

### Estimation of effective population size

The program *NeEstimator* v2.1 (*Do et al., 2014*) was used to estimate the effective population size ($N_E$) by applying the linkage disequilibrium method described in *Waples & Do (2010)*. This method relies on deviations from expected frequencies of gametes and genotypes to estimate $N_E$, calculating the linkage disequilibrium with the Burrow's $\Delta$

parameter. A random mating scenario was selected, and singletons were excluded from the analysis, since they contribute the most to an upward bias in the estimation of $N_E$ when using the linkage disequilibrium method (*Do et al., 2014*). A jackknife estimation of $N_E$ with a 95% confidence interval was implemented (*Jones, Ovenden & Wang, 2016*).

Since it has been shown that this method works better when using at least 30 individuals per population/location (*Nunziata & Weisrock, 2018*), we relaxed our average coverage threshold from 20× to 15× to include more individuals. Similarly, the minor allele frequency was reduced to 1% as increasing this filter could substantially decrease the $N_E$ estimation (*Marandel et al., 2020*). The function *filter_ld* from the R package *radiator* (*Gosselin et al., 2020*), which relies on tools from the package *SNPRelate* (*Zheng et al., 2012*), was run to correct for possible effects of linkage disequilibrium (*i.e.*, to make sure that one SNP was present on each read and to remove SNP's based on Long Distance Linkage Disequilibrium).

To ensure that we had only neutral markers, we removed loci responsible for the highest 5% $F_{ST}$ values across all samples following *Narum & Hess (2011)*. They showed with simulations that a 5% threshold correctly separated markers under strong selection, just as well as outlier tests using different programs. Note, however, that the efficacy of these methods is lower for models with weaker selection. We tested the effects of sub-sampling as suggested by *England et al. (2006)* and observed that most of the populations required a value of N = 20 to stabilize the estimations of $N_E$. Hence, we excluded locations with a smaller sample size (*i.e.*, Efra Sog, Villingavatnsá, Hengladalsá (Innstidalur), Hengladalsá (Fremstidalur), Hvítá, Ölfusá, Varmá).

### Detection of population bottleneck signals in Lake Þingvallavatn

Given the decrease in brown trout catches in the 1980s in Þingvallavatn, we tested for signals of a population bottleneck using the program *Bottleneck* v1.2.02 (*Cornuet & Luikart, 1996*). This program relies on the difference between rates of decrease in allele frequencies and general genetic diversity *sensu* Nei (*i.e.*, expected heterozygosity, $H_E$), the former being much faster immediately following a bottleneck event. Hence a population that experienced a recent bottleneck will exhibit an apparent heterozygosity excess (*Maruyama & Fuerst, 1985*; *Luikart et al., 1998*). The three available models in the program were run simultaneously: (i) infinite allele model (IAM), which adds a single mutation at a time until the number of alleles reaches the observed value; (ii) step-wise mutation model (SMM), which is a Bayesian approach where the likelihood of the parameter Θ (*i.e.*, $4N_E \mu$) is assessed as the proportion of iterations that yield exactly the number of alleles observed; (iii) two-phased model of mutation (TPM), which is intermediate to IAM and SMM and was run with a variance of 30 and a proportion of SMM set at 70%. Although originally designed for microsatellites, these models have been successfully applied to investigate bottlenecks using RADseq data in foxes, mosquitoes and serpentinophyte plants (*Funk et al., 2016*; *Maffey et al., 2020*; *Stojanova et al., 2020*).

To test for a significant heterozygosity excess relative to the equilibrium-expected value, a one-tailed Wilcoxon test was performed for three sets of 200 randomly selected loci per population (as defined per sampling location), since the program had trouble dealing with

a higher number of markers. Similarly, as no mutation rate estimates are available for brown trout, the SNP mutation rate of the lake whitefish *Coregonus clupeaformis* (*i.e.*, $1 \times 10^{-8}$) (*Rougeux, Bernatchez & Gagnaire, 2017*) was used.

The dataset used for the $N_E$ estimation was also used in the population bottleneck test (*i.e.*, using a 15× coverage threshold). However, this time the program *populations* was used to call haplotypes instead of single SNPs per locus. All subsequent filtering steps were run in a similar fashion (*e.g.*, minor allele frequency threshold of 0.05, removing highly related individuals). Establishing a minor allele frequency threshold of 5% is common practice and works under the assumption that the removed loci are not very informative at the population level. However, this strategy may be removing true rare alleles that could aid in the inference of a bottleneck (*O'Leary et al., 2018*). Nevertheless, we confirmed with a PCA comparison that this dataset gave the same cluster differentiation patterns as data with only one single SNP per locus (Fig. S2).

## RESULTS

### Genetic diversity

A total of 2,597 polymorphic loci from 317 individuals out of 555 originally analysed were retained after the stringent filtering steps outlined in Materials and Methods. These markers were used to address questions about the patterns of genetic differentiation between locations. We estimated population genetic parameters for each sampling site. Heterozygosity varied more than threefold, from 0.080 for Leirvogsvatn (LEI) and Þverá/upper Ölfusvatnsá (THV) to 0.289 for Hvítá (HVI) and Ölfusá (OLF) (Table S4). In most locations the observed heterozygosity was lower than expected, *i.e.*, $H_O$ *vs* $H_E$ in Table S4 ($p < 0.05$, Bartlett's test of homogeneity of variances).

The average nucleotide diversity also varied more than threefold among samples (Fig. 2). The putative anadromous populations and the Scottish reference group (Loch Slapin (SLP), also anadromous) had the highest nucleotide diversity. The individuals from Þingvallavatn (THI) and Úlfljótsvatn (ULF) as well as those from the rivers discharging into them, *i.e.*, Öxará (OXA), lower Ölfusvatnsá (FUS), Villingavatnsá (VIL), and Efra Sog (EFR) showed similar but notably lower levels (*i.e.*, π between $4.30$–$5.64 \times 10^{-4}$), but in Þverá/upper Ölfusvatnsá (THV), isolated from these locations by an impassable waterfall (see Fig. 1), the nucleotide diversity was even lower ($\pi = 3.26 \times 10^{-4}$). Similar low genetic diversity was seen in the headwater population of Leirvogsvatn (LEI) ($\pi = 2.58 \times 10^{-4}$) (Table S4). Interestingly, sampling sites from tributaries streams of Hengladalsá in three valleys, *i.e.*, from Innstidalur (HIN), Miðdalur (MID), and Fremstidalur (FRE) displayed intermediate levels of diversity, despite being isolated from the lowland locations by waterfalls (Fig. 2). The diversity levels were 2–3 times higher than in the other mountain population *i.e.*, Þverá/upper Ölfusvatnsá (THV). Notably, broader variance in diversity estimates was seen in locations that showed evidence of admixture, *i.e.*, Hestvatn (HES), Hengladalsá at the Miðdalur valley (MID), Ölfusá (OLF), Þingvallavatn (THI) and Úlfljótsvatn (ULF) samples.
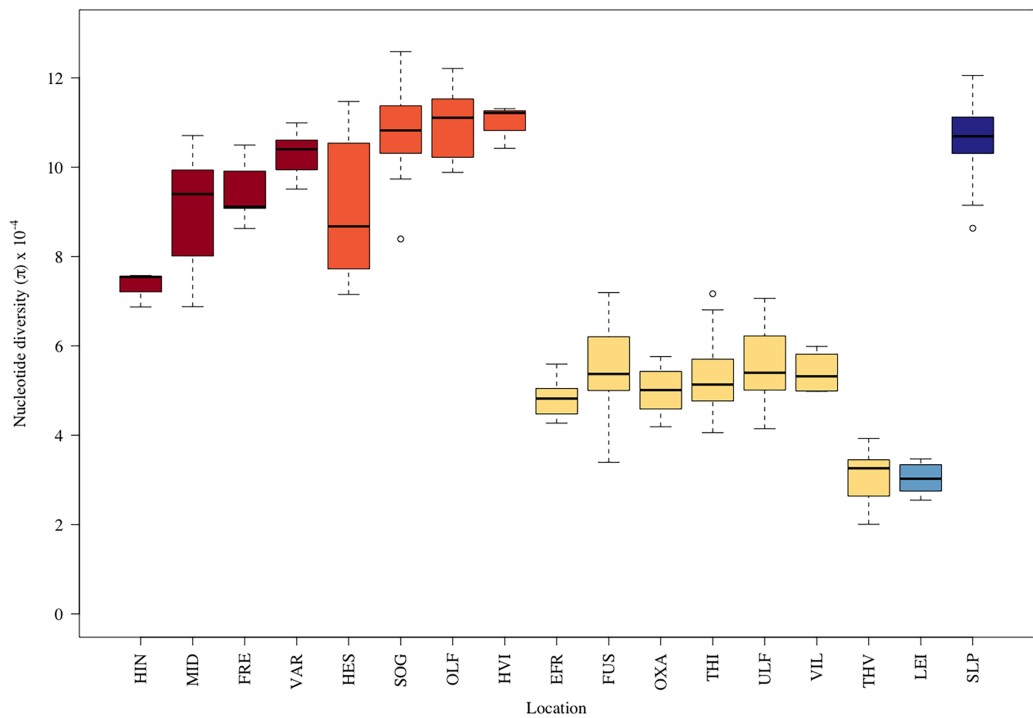

**Figure 2 Boxplots displaying the nucleotide diversity in the brown trout by sampling sites.** See Table 1 for three letter code.

## Structure and gene flow between brown trout populations

PCA, pairwise $F_{ST}$, admixture analyses, and a phylogenetic tree gave congruent results on genetic differentiation. Individuals from many of the sampled locations separated along the first two PC axes (Fig. 3). The first component separated the populations isolated above waterfalls into three distinct clusters, *i.e.*, the Þingvallavatn/Úlfljótsvatn watershed including Þverá/upper Ölfusvatnsá (THV), the southwest Hengill trout, *i.e.*, in tributary streams of Hengladalsá at Innstidalur (HIN), Miðdalur (MID), and Fremstidalur (FRE) valleys, and the headwater local reference population in Leirvogsvatn (LEI). The samples from the putative anadromous populations in the lowlands of Ölfusá (OLF), Hvítá (HVI), and Sog (SOG) as well as those from Hestvatn (HES), a lake connected to Hvítá (HVI), formed the fourth cluster, suggesting a close relation between brown trout from these sampling sites. Interestingly, the fish caught in Varmá (VAR), which we expected to group with the putative anadromous samples because of the connected waterways, clustered with the southwest Hengill populations. Also noteworthy is the fact that the Scottish reference group (SLP) grouped with Ölfusá (OLF) and Varmá (VAR) on the first axis but only separated from the Icelandic trout on the second axis. The third component separated three clusters, one composed of samples from Hestvatn (HES), one with individuals from Ölfusá (OLF), Sog (SOG), Hvítá (HVI), with a few individuals from Hestvatn (HES), and one with the samples from the rest of the locations. The samples from Leirvogsvatn (LEI) separated along the fourth principal component (Fig. S2).
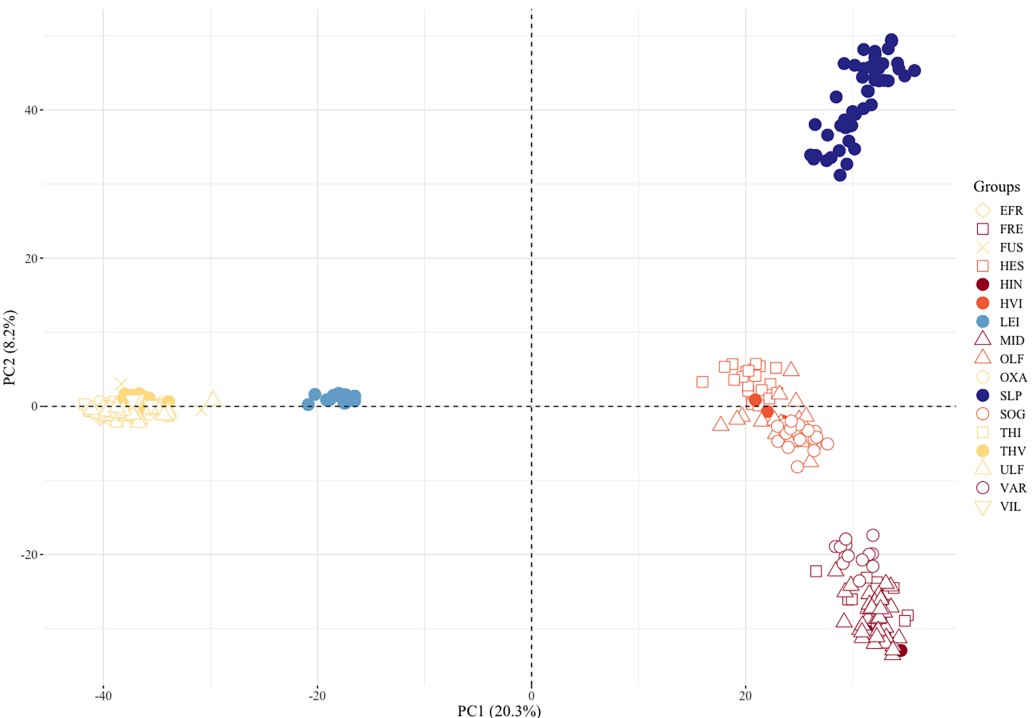

**Figure 3 Plot of two principal components separates three clusters of landlocked brown trout, the Scottish reference group, and the anadromous populations.** The percentage of the variance explained by these components is shown in parentheses. See Table 1 for three letter code.

The pairwise estimates of $F_{ST}$ showed the same patterns and quantified the differences between locations (Table 2). Consistently, the $F_{ST}$ values among the sample pairs from the Þingvallavatn/Úlfljótsvatn watershed were generally low ($F_{ST}$ ranging from 0.005–0.134), *i.e.*, group ABOVE (I) in Table 2, except for the isolated Þverá/upper Ölfusvatnsá (THV) population, which had an average $F_{ST}$ = 0.311. Similarly, fish in Hvítá (HVI), Sog (SOG), and Ölfusá (OLF) (*i.e.*, group BELOW in Table 2) had low pairwise $F_{ST}$ values (average $F_{ST}$ = 0.03 when compared among themselves) suggesting that they stem from the same population or belong to sub-populations that have interbred in the recent past. Varmá (VAR) and Hestvatn (HES) are connected to Ölfusá (OLF) and Hvítá (HVI) without any obvious physical barriers to fish migration. Yet the fish from Varmá (VAR) and Hestvatn (HES) deviated strongly from fish caught in Hvítá (HVI), Sog (SOG), and Ölfusá (OLF) (average $F_{ST}$ = 0.132 and 0.158, respectively).

The highest $F_{ST}$ values were observed between Leirvogsvatn (LEI) and the rest of the samples from the drainage basin of Ölfusá (*i.e.*, average $F_{ST}$ of 0.513), which curiously was almost two times higher than the average $F_{ST}$ (0.298) between the Scottish population (SLP) and the Icelandic samples/populations.

Finally, brown trout from Hengladalsá at Innstidalur (HIN) and Miðdalur (MID) were closely related (*i.e.*, $F_{ST}$ = 0.054) despite being separated by a waterfall, suggesting downstream gene-flow from the Innstidalur valley (HIN).

**Table 2 Pairwise $F_{ST}$ scores for the 17 brown trout locations featured, grouped by location.**

| | Above[1] | | | | | | | | | Below[2] | | | | | Other[3] | |
| | I | | | | | II | III | | | | | | | | | |
| | THI | FUS | VIL | EFR | ULF | THV | HIN | MID | FRE | VAR | HES | HVI | SOG | OLF | LEI | SLP |
|---|---|---|---|---|---|---|---|---|---|---|---|---|---|---|---|---|
| **OXA** | 0.033 | 0.047 | 0.134 | 0.012 | 0.015 | 0.359 | 0.602 | 0.461 | 0.504 | 0.478 | 0.449 | 0.498 | 0.426 | 0.411 | 0.510 | 0.413 |
| **THI** | – | 0.005 | 0.058 | 0.027 | 0.026 | 0.237 | 0.564 | 0.435 | 0.470 | 0.443 | 0.415 | 0.451 | 0.392 | 0.373 | 0.475 | 0.387 |
| **FUS** | | – | 0.059 | 0.049 | 0.032 | 0.227 | 0.566 | 0.435 | 0.470 | 0.443 | 0.416 | 0.452 | 0.392 | 0.372 | 0.479 | 0.386 |
| **VIL** | | | – | 0.131 | 0.100 | 0.322 | 0.563 | 0.425 | 0.460 | 0.432 | 0.408 | 0.447 | 0.383 | 0.363 | 0.482 | 0.379 |
| **EFR** | | | | – | 0.012 | 0.418 | 0.582 | 0.407 | 0.429 | 0.393 | 0.379 | 0.422 | 0.344 | 0.318 | 0.563 | 0.354 |
| **ULF** | | | | | – | 0.322 | 0.563 | 0.425 | 0.460 | 0.432 | 0.408 | 0.447 | 0.383 | 0.363 | 0.482 | 0.379 |
| **THV** | | | | | | – | 0.698 | 0.481 | 0.535 | 0.512 | 0.478 | 0.584 | 0.451 | 0.433 | 0.625 | 0.421 |
| **HIN** | | | | | | | – | 0.054 | 0.162 | 0.144 | 0.303 | 0.253 | 0.223 | 0.186 | 0.679 | 0.277 |
| **MID** | | | | | | | | – | 0.035 | 0.057 | 0.245 | 0.164 | 0.172 | 0.143 | 0.462 | 0.252 |
| **FRE** | | | | | | | | | – | 0.060 | 0.247 | 0.161 | 0.170 | 0.137 | 0.514 | 0.243 |
| **VAR** | | | | | | | | | | – | 0.206 | 0.107 | 0.125 | 0.092 | 0.489 | 0.210 |
| **HES** | | | | | | | | | | | – | 0.141 | 0.165 | 0.122 | 0.468 | 0.224 |
| **HVI** | | | | | | | | | | | | – | 0.060 | 0.000 | 0.562 | 0.147 |
| **SOG** | | | | | | | | | | | | | – | 0.032 | 0.436 | 0.188 |
| **OLF** | | | | | | | | | | | | | | – | 0.416 | 0.153 |
| **LEI** | | | | | | | | | | | | | | | – | 0.408 |

**Note:**

[1] The group ABOVE (I) refers to locations sampled in the watershed of Þingvallavatn and Úlfljótsvatn and above the Írafoss and Kistufoss waterfalls in Sog (*i.e.*, OXA, Öxará; THI, Þingvallavatn; FUS, lower Ölfusvatnsá; VIL, Villingavatnsá; EFR, Efra Sog; ULF, Úlfljótsvatn). The group ABOVE (II) consists of THV: Þverá/upper Ölfusvatnsá, located above a waterfall and flowing into Ölfusvatnsá and in turn, Þingvallavatn. The group ABOVE (III) includes samples from small tributaries in Hengladalsá in the southern slopes of Hengill and above impassable waterfalls in Hengladalsá (*i.e.*, HIN, Innstidalur; FRE, Fremstidalur; MID, Miðdalur).

[2] The group BELOW consists of locations in the lower reaches of the Ölfusá drainage basin and below the aforementioned waterfalls (*i.e.*, VAR, Varmá; HES, Hestvatn; HVI, Hvítá; SOG, Sog; OLF, Ölfusá).

[3] The group OTHERS includes Leirvogsvatn (LEI), which flows into the Atlantic ocean via Leirvogsá near Reykjavík and the Scottish reference group from Loch Slapin, in the Isle of Skye (SLP).

All values were significantly different from zero, except for the pair HVI-OLF.

The entropy criterion obtained in the admixture analysis reached a minimum at K = 8 (Fig. S3, see Fig. 4 for K = 6–8 and Fig. S4 for K = 9–11). Consistent with other analyses, the Scottish samples formed one cluster and the other Icelandic samples the seven remaining clusters. Brown trout from the tributary streams of Hengladalsá, *i.e.*, at Innstidalur (HIN), Miðdalur (MID), and Fremstidalur (FRE) valleys and downstream in Varmá (VAR) split into two groups (*i.e.*, upper and lower, C1 and C2 respectively), the samples from Hvítá (HVI), Sog (SOG), Ölfusá (OLF), and Hestvatn (HES) were grouped together in C3 while some individuals from Hestvatn (HES) split from the Hvítá (HVI) and Sog (SOG) cluster to form their own group (C4). Most fish caught in the watershed of Þingvallavatn and Úlfljótsvatn clustered together (C5), the exception being the fish from Þverá/upper Ölfusvatnsá (THV) plus a few fish caught in lower Ölfusvatnsá (FUS), Villingavatnsá (VIL) and Þingvallavatn (THI) that clustered separately (C6). Leirvogsvatn (LEI), and the Scottish reference population (SLP) formed distinct clusters, C7 and C8, respectively (Fig. 4).

The admixture proportions varied per location (K = 8, Fig. 5). The results clearly capture the contributions of the small headwater populations, *i.e.*, Innstidalur (HIN),

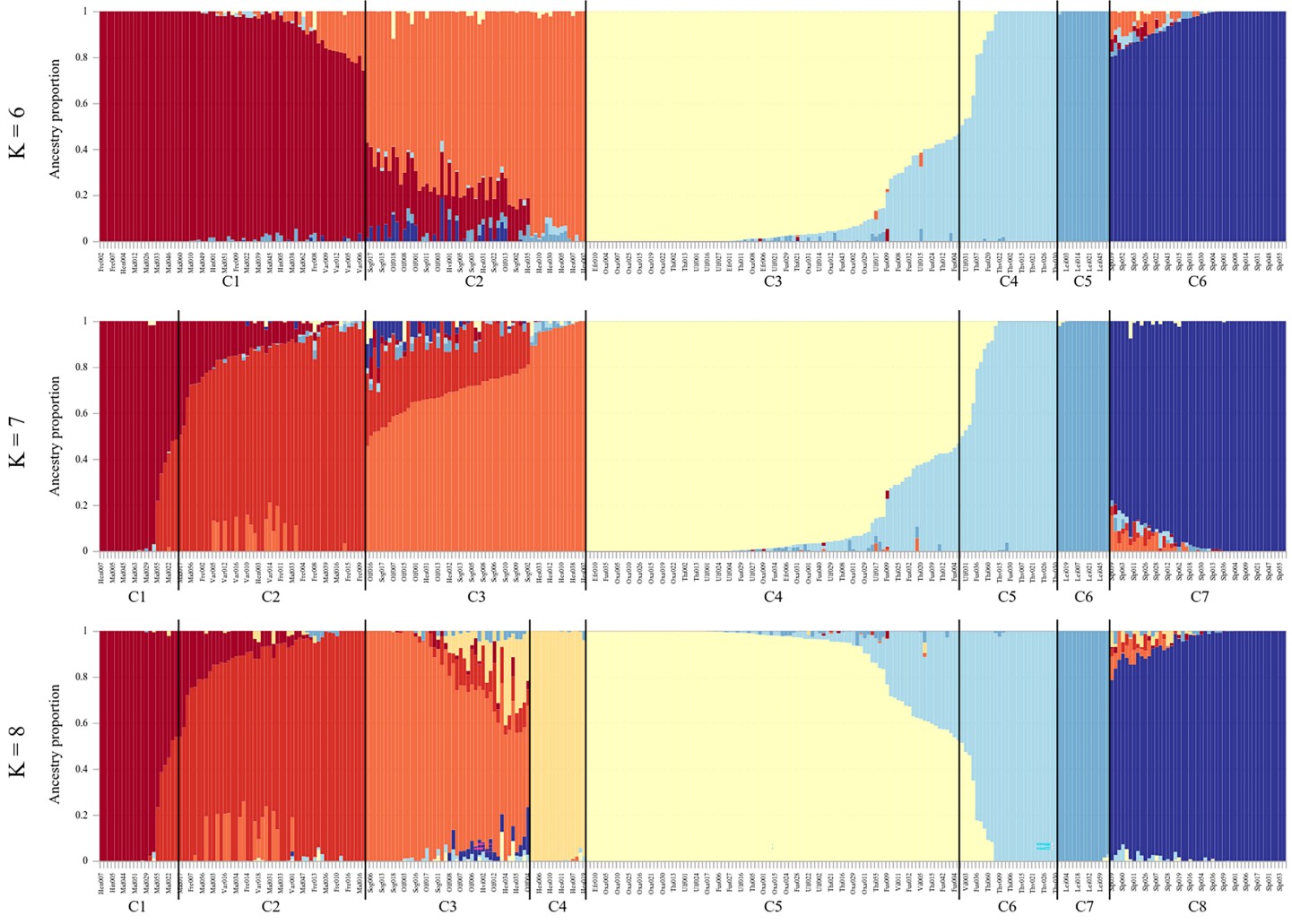

**Figure 4 Ancestry matrices for the sampled brown trout, with K = 6–8.** Note, not all samples from each location grouped in the same cluster. Using K = 8, C1, samples from the upper Hengladalsá including all from Innstidalur valley as well as some individuals from Miðdalur and Fremstidalur; C2, samples from the lower Hengladalsá including the valleys Miðdalur and Fremstidalur as well as Varmá (below waterfalls in Kambabrún); C3, individuals from Sog, Hvítá, Ölfusá and several from Hestvatn; C4, individuals from Hestvatn; C5, samples from Þingvallavatn, Öxará, lower Ölfusvatnsá, and Villingavatnsá as well as Efra Sog and Úlfljótsvatn; C6, samples from Þverá/upper Ölfusvatnsá and some from lower Ölfusvatnsá, Þingvallavatn, Villingavatnsá, and Úlfljótsvatn; C7, samples from Leirvogsvatn; C8, only individuals from the reference group from the Isle of Skye (Scotland).               

Þverá/upper Ölfusvatnsá (THV), and Hestvatn (HES) into downstream locations. The data do not indicate downstream flow of genes from the Þingvallavatn/Úlfljótsvatn system into the putative anadromous populations in the lowlands.

A unique genetic component from Þverá/upper Ölfusvatnsá (THV) appeared downstream in the fish caught in the lower Ölfusvatnsá (FUS) (directly below a waterfall) and Villingavatnsá (VIL). The latter river runs parallel to Ölfusvatnsá in the lower reaches and has its mouth in Þingvallavatn *ca.* 700 m to the south of the former (Fig. 1). This genetic component was present in most of the fish caught in Þingvallavatn (THI) and to a lesser extent, in the fish from Úlfljótsvatn (ULF) (Fig. 5), but notably not in fish caught spawning in Öxará (OXA).

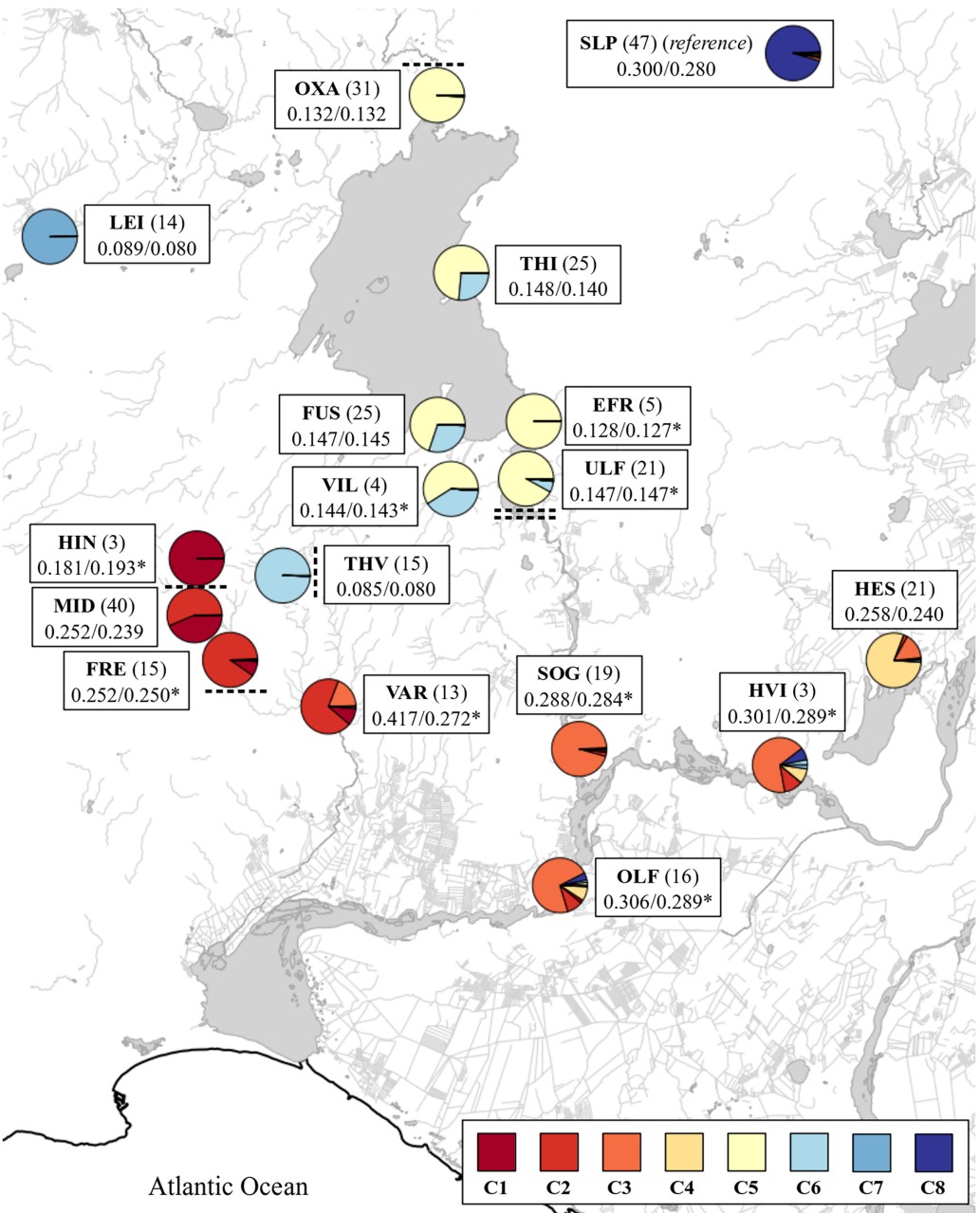

**Figure 5 Map with admixture pie plots using the number of ancestral populations K = 8.** The cluster names in the inset correspond to those from Fig. 4. See Table 1 for three letter code. Sample size is indicated in parenthesis. Expected and observed heterozygosities ($H_E$ and $H_O$) are separated by a forward slash. A asterisk (*) indicates significant differences between observed and expected values ($p < 0.05$, Bartlett test of homogeneity of variances). Waterfalls above and below sampling sites are indicated with dashed lines.

There was clear evidence of downstream gene flow from the populations inhabiting the streams on the south side of the Hengill volcano. Again, Hengladalsá at Innstidalur (HIN) and Miðdalur (MID) were closely related (*i.e.*, $F_{ST}$ = 0.054) despite being separated by a waterfall, and the ancestry plots suggested a downstream flow of genetic material from

Innstidalur (HIN). The samples from the upper parts of Hengladalsá (*i.e.*, at the Innstidalur valley, HIN) showed a unique component that progressively decreased downstream. The samples from Varmá (VAR) clustered with individuals from Hengladalsá (Fig. 3) and the ancestry plot showed a high proportion of a genetic component from Henglandalsá (Fig. 4). This component also appeared in putative anadromous fish in the lowlands, *i.e.*, Hvítá (HVI), Sog (SOG), Ölfusá (OLF) (Fig. 4). Note also that the fish caught in Varmá (VAR) have a significant genetic contribution from Hvítá (HVI)/Sog (SOG)/Ölfusá (OLF) (Table S5), reflecting the lack of barriers to upward migration from Ölfusá (OLF).

Mixing or contribution of genes between Hestvatn (HES) and the anadromous fish in Hvítá (HVI), Sog (SOG), and Ölfusá (OLF) was also evident, but the data also indicate that some Hestvatn (HES) trout belong to a resident population. A genetic component from the lowlands, which was most predominant in the Sog (SOG) samples, appeared in some individuals caught in Hestvatn (HES).

A phylogenetic analysis with *RAxML* while largely congruent with the PCA revealed several extra observations (Fig. 6). The samples from the Þingvallavatn/Úlfljótsvatn watershed, *i.e.*, Þingvallavatn (THI), Öxará (OXA), lower Ölfusvatnsá (FUS), Þverá/upper Ölfusvatnsá (THV), Villingavatnsá (VIL), Efra Sog (EFR), Úlfljótsvatn (ULF) and those from the south-western slopes of Hengill, *i.e.*, Varmá (VAR), Fremstidalur (FRE), Miðdalur (MID), Innstidalur (HIN) formed two distinct clades separated by a large phylogenetic distance. It should be noted that the brown trout in the southwestern slope of Hengill and the river Þverá are relatively close geographically (*ca*. 1 km apart, separated by a mountain ridge) but the streams drain in opposite directions.

The headwater population of Leirvogsvatn (LEI) and the reference group from Scotland (SLP) formed their own distinct clades (Fig. 6). A well-supported branch containing Leirvogsvatn (LEI) was placed near the Þingvallavatn/Úlfljótsvatn branch (Fig. 1).

The clade distribution of individuals from the lowland putative anadromous populations was less clear, with individuals from Sog (SOG), Ölfusá (OLF), and Hvítá (HVI) landing on different poorly separated branches. Samples from Hestvatn (HES) grouped in two clusters on the same well supported branch. The samples from Varmá (VAR) connected to the stem of the well supported Hvítá (HVI)/Sog (SOG)/Ölfusá (OLF) branch. Samples upstream of Varmá (VAR) formed a well-supported clade containing a mix of individuals from the lower parts of Hengladalsá (*i.e.*, Miðdalur (MID) and Fremstidalur (FRE) valleys) and a sub-branch with individuals from the upper part of the river, *i.e.*, Innstidalur (HIN) as well as some from Miðdalur (MID).

## Effective population size and test for a population bottleneck in Þingvallavatn

We wanted to estimate the effective size of the populations sampled at each location. The filtering approach designed to retain more individuals and markers yielded 377 individuals and 5,353 polymorphic loci, but the LD correction reduced this to 3,505 markers (Table S6) All sampled locations showed low effective population sizes, with the smallest value in Leirvogsvatn (LEI) (*i.e.*, $N_E = 3.4$) (Table 3). Brown trout from Sog (SOG),

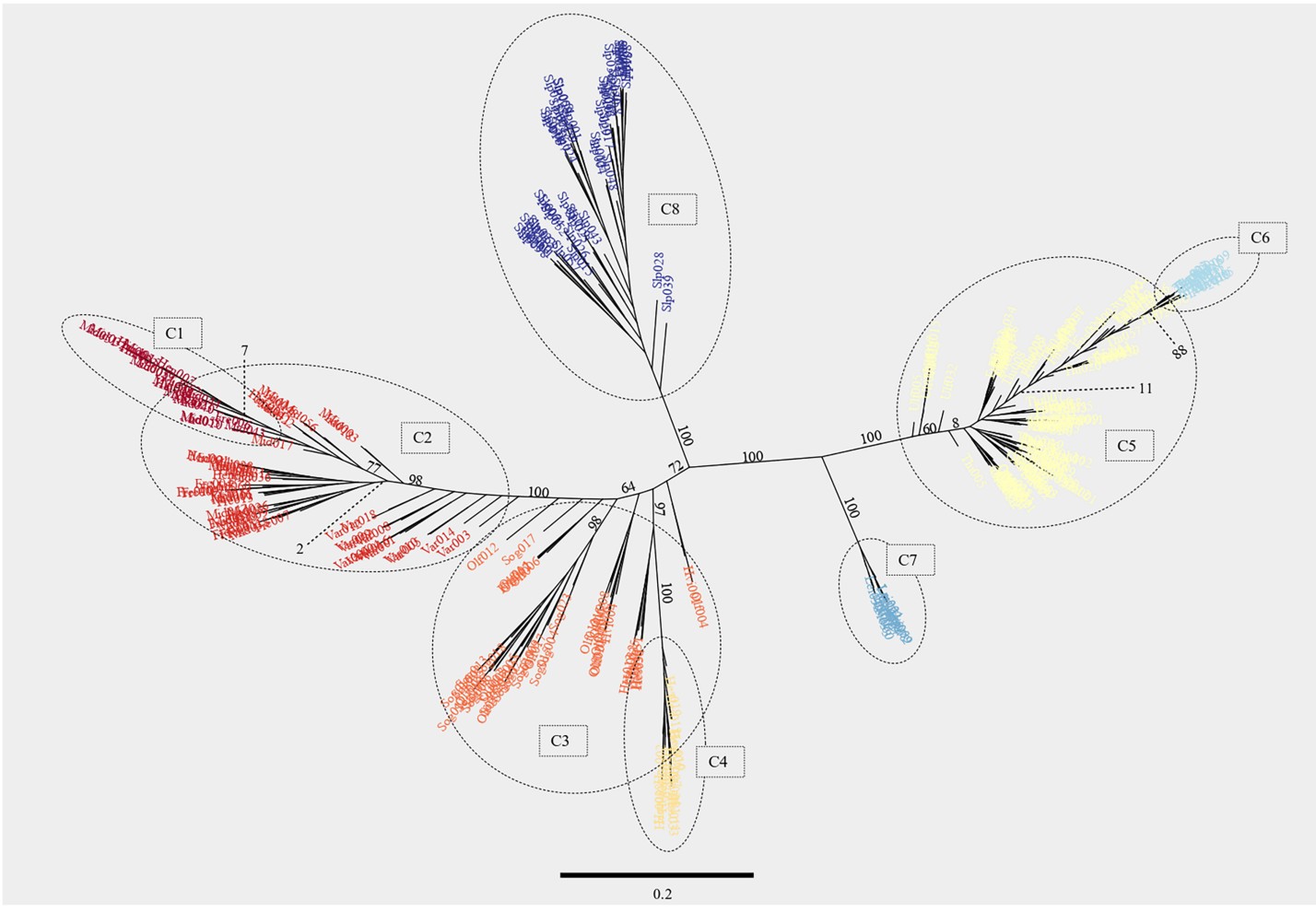

**Figure 6 Maximum likelihood phylogenetic tree generated with the software *RAxML* showing individuals from the 17 locations studied.** The model used in this analysis was GTR-GAMMA, which is a version of the General Time Reversible model of nucleotide substitution under the Gamma model of rate heterogeneity. After 550 iterations of bootstrapping the topology of the tree converged to the one shown here. The numbers on branches indicate the support values (percentages). The scale bar represents the amount of genetic change between sequences associated with branch lengths (*i.e.*, 0.1 represents a 10% difference between sequences per length of branch). The support for a few selected branches at the base of clusters is shown with dotted lines. The eight clusters obtained in the ancestry proportion analysis with K = 8 were delimited with broken lines. Individuals belonging to each cluster reflect the color palette from Fig. 4. See Table 1 for three letter code.

presumably anadromous fish, had the highest effective population size (*i.e.*, $N_E$ = 71.2). Notably, the estimate for Öxará (OXA) was the second largest (*i.e.*, $N_E$ = 56.5) while that of lower Ölfusvatnsá (FUS), showed an $N_E$ = 14.9 (Table 3). Recall, both drain into Þingvallavatn and clustered together by genetic relatedness.

To test for a bottleneck in the population of Þingvallavatn, we used a different dataset (*i.e.*, 3,019 polymorphic loci, 373 individuals) using haplotypes and excluding sampling locations with <20 individuals. For reference, other locations with *n* > 20, including the Scottish reference group, were also tested.

None of the three runs for the three different models (*i.e.*, IAM, SMM, TPM) yielded significant results for the Wilcoxon tests for heterozygosity excess, except for the reference

**Table 3 Estimates of effective population size for sites with 20 or more individuals sampled.**

| Code | N | $N_E$ | CI 2.5% | CI 97.5% |
| --- | --- | --- | --- | --- |
| FUS | 23 | 14.9 | 8.5 | 30.9 |
| OXA | 31 | 56.5 | 42.1 | 82.9 |
| ULF | 24 | 21.0 | 13.0 | 40.0 |
| THV | 23 | 14.7 | 9.7 | 24.3 |
| MID | 38 | 14.3 | 9.5 | 22.0 |
| HES | 30 | 10.5 | 7.5 | 14.7 |
| SOG | 20 | 71.2 | 41.2 | 209.3 |
| LEI | 20 | 3.4 | 2.7 | 6.3 |
| SLP | 48 | 44.7 | 34.2 | 61.6 |

**Note:**

N, sample size; $N_E$, effective population size calculated using the method based on linkage disequilibrium and 95% non-parametric jackknife confidence limits. The samples caught in Þingvallavatn were removed because the origin of these fish is not known, as they likely represent several spawning populations (*e.g.*, Öxará, lower Ölfusvatnsá, Villingavatnsá).

group SLP with the IAM model. All other sampling locations showed significant heterozygote deficiency ($p < 0.001$).

# DISCUSSION

We set out our study to assess the genetic structure of brown trout populations in the Ölfusá watershed for the first time and to answer questions related to their diversity, connectivity, and effective population size by using SNP's as genetic markers obtained from ddRADseq data. The results show that the genetic structure of brown trout in the western part of the Ölfusá watershed largely reflect the present connectivity and topography of the waterways as well as the timing by which important features of the waterways were formed, *e.g.*, impassable waterfalls. Here we discuss population structure, genetic diversity, and effective population size with emphasis on the brown trout of lake Þingvallavatn.

## Structure and gene flow between brown trout populations

The results ($F_{ST}$ values, PCA and the phylogenetic tree) showed a pattern and degree of genetic structure that fit expectations based on the present topography and connectedness of waterways in the western part of the Ölfusá watershed. The headwater populations, which are located above impassable waterfalls, showed clear signs of isolation from the putative anadromous lowland populations, which may exchange genetic material more freely. Importantly, individuals from Úlfljótsvatn, Þingvallavatn, and its tributaries, *i.e.*, Úlfljótsvatn (ULF), Þingvallavatn (THI), Öxará (OXA), Ölfusvatnsá (FUS), Villingavatnsá (VIL), and Efra Sog (EFR) clustered together and most of their pairwise $F_{ST}$ values were low (range 0.005–0.134), implying that the populations of these lakes and tributaries are genetically very similar. As was expected, the Þverá/upper Ölfusvatnsá (THV) population, located above an impassable waterfall, was the only one forming a genetically distinct sub-group within this clade. The significant ancestry component from this population in lower Ölfusvatnsá (FUS), Villingavatnsá (VIL), and Þingvallavatn (THI), suggests

significant downstream gene flow from Þverá/upper Ölfusvatnsá (THV). The observed pattern of admixture may stem from remnants of spawning populations in lower Ölfusvatnsá and Villingavatnsá, which, due to geographic proximity, would be affected by down-stream gene flow from the Þverá/upper Ölfusvatnsá (THV) population, an influence not seen in the Öxará spawners.

Although written records of brown trout spawning sites are scant, it is generally thought that prior to the construction of the dam the trout of Þingvallavatn originated from several sub-populations of spawners. The best-known spawning locations are in the rivers Öxará, lower Ölfusvatnsá, Villingavatnsá, and close to and in the outlet at Efra Sog (*Jóhannsson & Jónsson, 2002*) and according to farmers around the lake, trout was also known to spawn at several cold-water spring sites within the lake (*Skarphéðinsson, 1996*). The lack of a subgroup structure reflected by the low $F_{ST}$ values among the fish caught in these locations and the dominant admixture component from the Öxará population (OXA) seen at present may to a large extent be a consequence of the efforts to reinvigorate the trout of Þingvallavatn.

It is notable that the parr caught in Villingavatnsá (VIL), which had no releases of fingerlings from Öxará (OXA), were the most genetically distinct among the samples from Þingvallavatn, Úlfljótsvatn and its tributaries ($F_{ST}$ values ranging from 0.058–0.134). Moreover, they show the most definite genetic component from the isolated population in Þverá/upper Ölfusvatnsá (THV). The fish caught in lower Ölfusvatnsá (FUS) were more mixed in this respect, some showing a strong component from Þverá/upper-Ölfusvatnsá (THV) while others mostly had the Öxará (OXA) component. This suggests that the releases of parr and planting of eggs from the Öxará spawning population lead to mixing of these components in the trout presently spawning in lower Ölfusvatnsá (FUS).

The planting of eggs and releasing of juveniles may have been crucial, as the hatched fish would have experienced the imprinting of the natal river conditions. As the Þverá/upper Ölfusvatnsá (THV) population consists of small trout, but the lake fish can reach very large sizes, it remains an open question whether this gene flow influences traits in individuals of mixed ancestry. When some Irish lakes were stocked with adults for a period of 7 years, the genetic material of the native populations remained intact because the introduced fish did not spawn in the streams, as they probably lacked the imprinting of the conditions of the natal streams at an early age (*O'Grady, 1984*).

Similar patterns of phylogenetic separation, with directional gene flow, were seen in mountain tributaries of Hengladalsá. These trout clustered on a well-supported branch (98%) of the phylogenetic tree. Although we only had three samples from the upper Hengladalsá (*i.e.*, from the Innstidalur valley (HIN), isolated above impassable waterfalls), the data suggest that these isolated tributaries harbour a genetically distinct population. Below these waterfalls, Miðdalur (MID) and Fremstidalur (FRE) had 15/40 and 1/15 fish with an Innstidalur (HIN) ancestry proportion >0.5, respectively. Most fish caught in the tributary streams of Miðdalur (MID) (25/40) and Fremstidalur (FRE) (14/15) showed a different genetic component that, somewhat unexpectedly, was found dominant in the fish from Varmá (VAR). Notably, this component was also found in trout from the Hvítá (HVI)/Sog (SOG)/Ölfusá (OLF) system. As might be expected, all the fish from Varmá

(VAR) had a small but significant ancestry proportion (0.08–0.25) from the Hvítá (HVI)/Sog (SOG)/Ölfusá (OLF) system. The reasons for the shift in genetic composition from Innstidalur (HIN) to Miðdalur (MID) are unclear but we hypothesize that this may be caused by differences in habitat. The Innstidalur (HIN) habitat only consists of cold springs whereas the fish from the Miðdalur (MID) and Fremstidalur (FRE) valleys were caught in streams influenced by geothermal activity which drastically changes the annual temperature regime and ecological cycles of these streams (*O'Gorman et al., 2016*; *Ólafsson, 2019*). It seems plausible that adaptation to this environment may have led to genetic differentiation of these populations. This could also explain the dominance of the Miðdalur (MID) genetic component in the fish from Varmá (VAR) as this river is also significantly influenced by geothermal springs.

Considering the historically large trout populations of the Þingvallavatn/Úlfljótsvatn system (*Skarphéðinsson, 1996*) it is somewhat surprising not to see some genetic contributions from these populations in the putative anadromous populations downstream of the Írafoss and Kistufoss waterfalls. While it seems inevitable that brown trout would accidentally be swept downstream of the waterfalls, such escapees do not mix effectively with downstream populations.

It is notable that fish from Leirvogsvatn (LEI), despite having the highest $F_{ST}$ values with respect to the samples from the Þingvallavatn/Úlfljótsvatn watershed (*i.e.*, OXA, THI, ULF, FUS, EFR, THV), are close to the Þingvallavatn/Úlfljótsvatn fish on the first PCA axis and are placed together, on a well-supported branch, in the phylogenetic tree. This suggests a common origin and early split between these populations. Presumably the early isolation of trout in the Þingvallavatn/Úlfljótsvatn system resulted in strong selection against downstream migration and rapid adaptation to a resident lifestyle, a process that would have been greatly enhanced by high productivity of benthic invertebrate prey and opportunities for switching to piscivory in these lakes. Migratory behaviour is a quantitative trait with a genetic component (*Palm & Ryman, 1999*; *Kendall et al., 2015*). A highly productive freshwater environment has also been linked to a reduced likelihood of anadromy (*Finstad & Hein, 2012*). Similarly, our data indicate that the early establishment of residency of the brown trout in Þingvallavatn and Úlfljótsvatn has contributed significantly to the high degree of genetic differentiation observed between them and the populations of the Hvítá/Sog/Ölfusá system.

Although fish from the connected lowland waterways in Hestvatn (HES), Hvítá (HVI), Sog (SOG), and Ölfusá (OLF) scattered on the same branch of the tree, most individuals caught in the same water body tended to cluster together on well supported sub-branches. In Hestvatn (HES), most of the fish clustered on a well-supported (100%) sub-branch while a few fish had a high proportion (around 50%) of a genetic component from the Hvítá (HVI)/Sog (SOG)/Ölfusá (OLF) system. Some individuals caught in Hestvatn had *ca*. 30% of the genetic component from the lake and about 50% of the Hvítá/Sog/Ölfusá component (Table S5), indicating genetic admixture. This suggests that Hestvatn harbours two sub-populations, a distinct resident population and migrants from River Hvítá or trout from interbreeding between those stocks. The stream connecting Hestvatn to Hvítá is less than 2 km long and there is no impassable obstacle that could prevent migration under

normal circumstances (M. Jóhannsson, 2021, personal communication). During floods in Hvítá glacial water flushes into Hestvatn. According to a farmer by the lake, such floods can bring spurts of migrating Atlantic salmon, Arctic charr and brown trout into Hestvatn (S. Reynisdóttir, 2021, personal communication).

## Small effective population size, and low genetic diversity

The $N_E$ estimates for several locations in the watershed of Ölfusá showed small values, the exception being the spawning trout from Öxará (OXA), and the fish sampled in Sog (SOG) which had larger $N_E$, and the population in Leirvogsvatn (LEI) with a very small $N_E$.

The observed variation in $N_E$ values may reflect habitat constrains that limit the number of sexually mature individuals coming to the spawning grounds.

Although we have no knowledge of spawners or spawning locations in the mountain streams of Hengill, it seems plausible that the adult population size could be limited by the availability of food in the small habitats of the mountain streams flowing into Hengladalsá (*i.e.*, HIN, MID, FRE) and Þverá/upper Ölfusvatnsá (THV), resulting in the small $N_E$ values observed. This is in stark contrast to the situation in Þingvallavatn, which offers vast habitats and favourable conditions for growing immature fish. Presently, several thousand individuals enter Öxará each year to spawn (*ca*. 3,000 in 2019, see *Laxfiskar, 2022*). Although the $N_E$ observed in this population (OXA) was almost four times larger than that of Þverá/upper Ölfusvatnsá (THV), it seems low considering the number of spawners. This strongly suggests a different type of habitat constraint, *i.e.*, that $N_E$ is limited by the size of the spawning area. The number and quality of suitable spawning areas can have a direct impact on the effective population size (*Heggenes et al., 2009*). For females, the competition for the chance to reproduce is associated with an increased risk of nest overlapping and embryo mortality because of delayed spawning (*Belmar-Lucero et al., 2012*). It seems most likely that the disparity between the $N_E$ estimate of the Öxará population and the number of spawners results from fierce competition and nest overlap in a limited spawning area, a situation that results in an outcome where only a small proportion of spawners are successful in leaving viable offspring.

Long isolated populations in small habitats with low carrying capacity are prone to have low effective population sizes (*Funk et al., 2016*). The small $N_E$ values observed in our study are in line with the literature. Small isolated brown trout populations showed $N_E < 100$ in central Italy streams (*Splendiani et al., 2019*; *Rossi et al., 2022*), northern Spain rivers (*González-Ferreras et al., 2022*), one oligotrophic lake in Sweden (*Charlier, Laikre & Ryman, 2012*), and in a metapopulation of a Swedish lake system (*Andersson et al., 2022*). *Linløkken, Johansen & Wilson (2014)* found $N_E$ values between 9.2–1,222.3 that correlated positively with size habitat in brown trout populations of a lake watershed in south-east Norway.

It is important to consider that the linkage disequilibrium method used to estimate $N_E$ may not only reflect the $N_E$ per generation, but also the number of breeding individuals per year (Nb), or something in between. For a species like brown trout (with overlapping generations and repeated spawning), Nb is expected to be lower than $N_E$ (*Waples et al., 2013*). In general, Nb in brown trout populations has been shown to be significantly lower

than $N_E$ (*Charlier, Laikre & Ryman, 2012*; *Serbezov et al., 2012*). Furthermore, a downward deviation of the $N_E$ estimation may be caused by genetic admixture (*Waples, 2006*), the presence of siblings (*Whiteley et al., 2013*; *Linløkken, Johansen & Wilson, 2014*), or loci under selection (*Linløkken, Johansen & Wilson, 2014*), while the presence of immigrants could cause an upward bias (*Serbezov et al., 2012*). Although we attempted to correct for some of these factors, the possibility of a downward bias remains.

The small effective population size values observed in this study may pose a threat to some of these populations. An effective population size in the range of 1,000–5,000 seems to be an adequate number to maintain the genetic diversity of populations in the long term (*Lynch & Lande, 1998*). It has been proposed that $N_E$ of 500 may be enough (*Frankham, 1995*; *Andersson et al., 2022*), which is a value far above all the estimates in this study. Populations with small effective population size have a reduced genetic diversity that compromises their adaptability (*Frankham, 1996*).

Our results revealed a clear pattern of genetic diversity by geography and waterway topography. The small headwater populations of Þverá/upper Ölfusvatnsá (THV) and Leirvogsvatn (LEI) showed the lowest diversity of the populations studied, with the average nucleotide diversity being *ca*. 70% lower than that of the putative anadromous lowland populations. The nucleotide diversity was significantly higher in samples from Úlfljótsvatn (ULF) and Þingvallavatn (THI) and the accessible part of its tributary streams but *ca*. 50% lower than that of the fish caught in the lowland rivers below the Írafoss and Kistufoss waterfalls.

The low values of observed heterozygosity ($H_O$) and nucleotide diversity estimates ($\pi$) in most of the headwater locations sampled are probably the result of many generations in isolation. However, in the case of Leirvogsvatn (LEI) and Þverá/upper Ölfusvatnsá (THV), habitat constraints could also play a role, *e.g.*, restricted spawning habitats (in Leirvogsvatn, LEI) and restricted winter habitat and adaptation to the ecological effects of the thermal springs (in Þverá/upper Ölfusvatnsá, THV). Considering the low levels of genetic diversity in Leirvogsvatn (LEI), Þverá/upper Ölfusvatnsá (THV), and the populations of Úlfljótsvatn (ULF) and Þingvallavatn (THI) and its tributaries, the relatively higher genetic diversity in the mountain streams in Hengladalsá is odd. This may reflect admixture caused by recent stocking or population substructure. The six fish from Innstidalur (HIN) were sampled in cold springs but the fish from Miðdalur (MID) and Fremstidalur (FRE) were sampled in several very small, thermally influenced streams (*O'Gorman et al., 2016*; *Ólafsson, 2019*) that enter the much colder Hengladalsá separately. Pooling small samples from these thermally diverse habitats possibly masks a hidden population structure thus inflating estimates of genetic diversity. An alternative, but not exclusive scenario, is the possibility of active releases of brown trout from lowland populations into Hengladalsá. *Guðjónsson, Guðmundsson & Jónsson (1983)* reported the release of brown trout in Varmá and/or Hengladalsá in the early 1960's. These records are unclear, however, as there is no indication whether this was done below and/or above the impassable waterfalls in Hengladalsá. Another factor to consider when analysing the values of $H_O$ and $\pi$ is that the removal of loci that deviate from Hardy-Weinberg equilibrium, as was done in this study, is known to reduce $H_O$ and $\pi$ estimates (*Shafer et al., 2017*).

However, since this was applied equally to all populations, the relative differences in genetic diversity should have been preserved.

### Did the dam and restoration efforts influence Lake Þingvallavatn brown trout?

Despite the observed drop in catches of trout in Þingvallavatn after the dam and power plant construction (*Malmquist, Snorrason & Skúlason, 1985*) and a sustained period (1970–1990) of very low catches and poor recruitment, our data showed no signs of a recent population bottleneck in the Þingvallavatn/Úlfljótsvatn watershed. While the severe drop in catches of brown trout in most areas of Þingvallavatn in the 1960's and 1970's may have been mostly caused by the loss of the largest spawning ground during the dam construction, it is difficult to explain how this could have caused the prolonged lows in recruitment at other spawning grounds. The history of trout fishing in Þingvallavatn before the dam suggests sub-populations of trout in Þingvallavatn may have fluctuated considerably over time, in some cases due to heavy fishing (*Skarphéðinsson, 1996*). Long-time (decadal) fluctuations in temperature and a steady global warming trend may also have played a role in these fluctuations. In northern latitudes global warming appears to favour brown trout relative to Arctic charr (*Svenning et al., 2022*) and this may explain the explosive growth of the trout population in Þingvallavatn in the last two decades (Q. J. B. Horta-Lacueva, F. Finn, F. Ingimarsson, H. R. Ingvarson, H. Xiao, K. H. Kapralova, L. Ponsioen, M. Lagunas, M. De La Camara, N. Eskafi, S. M. Stefánsson, S. S. Snorrason, 2022, unpublished data).

The stocking efforts in Þingvallavatn seem to have been successful despite some differences in the physical characteristics of rivers where the eggs originated, *i.e.*, Öxará, and where they were planted, *i.e.*, in the lower Ölfusvatnsá and near the outlet in Efra Sog (*Jóhannsson & Jónsson, 2000a*; *Jóhannsson & Jónsson, 2016b*; Q. J. B. Horta-Lacueva, F. Finn, F. Ingimarsson, H. R. Ingvarson, H. Xiao, K. H. Kapralova, L. Ponsioen, M. Lagunas, M. De La Camara, N. Eskafi, S. M. Stefánsson, S. S. Snorrason, 2022, unpublished data). Recruitment of juvenile trout from spawners in the lower Ölfusvatnsá seems plentiful at present (Q. J. B. Horta-Lacueva, F. Finn, F. Ingimarsson, H. R. Ingvarson, H. Xiao, K. H. Kapralova, L. Ponsioen, M. Lagunas, M. De La Camara, N. Eskafi, S. M. Stefánsson, S. S. Snorrason, 2022, unpublished data). There can be little doubt that the planting of embryos and release of juveniles derived from the Öxará spawners has affected the genetic structure of the meta-population of trout in the Þingvallavatn/Úlfljótsvatn system. This is reflected in the strong genetic component from the Öxará spawners seen in samples from all other locations.

## CONCLUSION

Brown trout is a key player in the ecosystem of the Ölfusá watershed, and the identification of unique populations and genetic relations among populations is an important tool for proper management and conservation in this area.

Our results revealed the presence of seven genetically distinct clusters whose distinctness and links to other clusters, to a large extent, reflected the connectivity of the

waterways and topography of the watercourses. Some of these unique clusters represent populations in small water bodies where isolation and habitat constraints have led to reduced genetic diversity, *i.e.*, Leirvogsvatn (LEI), Þverá/upper Hengladalsá (THV). Despite small population sizes, we did see evidence of downstream gene flow from the small streams in the Hengill volcano, but notably, no such signals were seen in populations downstream of the much larger Þingvallavatn/Úlfljótsvatn system. Most sampling locations displayed rather small effective population sizes, except for Sog (SOG) and Öxará (OXA). Only two of eight populations measured showed a $N_E > 50$, which is a reason for concern (*Andersson et al., 2022*). The data did not indicate recent population bottlenecks in the brown trout of Þingvallavatn and hint at the presence of a metapopulation in the watershed of lakes Þingvallavatn and Úlfljótsvatn. Albeit the restoration efforts in the last century may have increased the contribution of the Öxará genetic component in the south of the lake. The preservation of spawning sites may be as important as the possibility of continued gene flow between the local sub-populations to maintain their current genetic diversity.

The destruction of spawning sites and overfishing are the main threats to genetic diversity in brown trout all over the world (*Ryman & Ståhl, 1981*; *Ferguson, 1989*). Both factors likely played a role in the decline of the Þingvallavatn trout in the 1960's and 1970's. Sharp cooling of the Icelandic climate in the late 1960's may have exacerbated the situation. The long-term survival of brown trout populations depends on effective management and protection of their habitat thereby ensuring the conservation of their genetic variability. Since it may be impossible to preserve all populations, identifying key populations that possess most of the genetic diversity in the area may be necessary before taking actions (*Ferguson et al., 1995*). *Swatdipong et al. (2009)* proposed a system of categorization of populations according to not only their genetic diversity but also their evolutionary distinctness from both a phenotypic and genotypic perspective. Perhaps such a strategy would help in the prioritization of a few populations in the Ölfusá watershed for conservation purposes.

Considering the populations featured in this study, we recommend that the isolated populations in the slopes of the Hengill volcano should be protected. Despite their small population size our data suggest that they contribute significant genetic variation to downstream populations in Varmá (VAR) and Þingvallavatn (THI). This could be important as these fish may harbour variation related to a wider tolerance of increased temperature.

Although the iconic Þingvallavatn trout seems to have made a drastic recovery in the last decade, better knowledge is needed to understand how its sub-populations in Öxará (OXA), Ölfusvatnsá (FUS), and Villingavatnsá (VIL), contribute to the metapopulation of trout in Þingvallavatn and Úlfljótsvatn.

## ACKNOWLEDGEMENTS

We would like to thank past and present members of the Arctic Charr Research Group at the University of Iceland for their help with sampling and helpful discussions. Sigrún Reynisdóttir helped with sampling in Hestvatn and Árni Guðmundsson provided the

samples from Hvítá. Jóhannes Guðbrandsson, Ian M. Dworkin, Völundur Hafstad and Gísli Már Gíslason helped with sampling in Hengill streams. We thank Jóhannes Sturlaugson for providing the samples from Öxará. Special thanks go to Jóhann Jónsson and the late Rósa Jónsdóttir, farmers at Mjóanes, for their kind help in the sampling of the trout of Þingvallavatn.

### Funding

This work was supported with a Landsvirkjun Research Grant and a University of Iceland Doctoral Research Grant (project number 1535-1533098). The funders had no role in study design, data collection and analysis, decision to publish, or preparation of the manuscript.

### Grant Disclosures

The following grant information was disclosed by the authors:
Landsvirkjun Research Grant.
University of Iceland Doctoral Research Grant: 1535-1533098.

### Competing Interests

The authors declare that they have no competing interests.

### Author Contributions

- Marcos Lagunas conceived and designed the sampling scheme, performed the sampling and laboratory work, analyzed the data, prepared figures and/or tables, authored or reviewed drafts of the article, and approved the final draft.
- Arnar Pálsson conceived and designed the sampling scheme, performed the sampling, authored or reviewed drafts of the article, and approved the final draft.
- Benóný Jónsson performed the sampling, authored or reviewed drafts of the article, and approved the final draft.
- Magnús Jóhannsson performed the sampling, authored or reviewed drafts of the article, and approved the final draft.
- Zophonías O. Jónsson conceived and designed the sampling scheme, performed the sampling, authored or reviewed drafts of the article, and approved the final draft.
- Sigurður S. Snorrason conceived and designed the sampling scheme, performed the sampling, authored or reviewed drafts of the article, and approved the final draft.

### Animal Ethics

The following information was supplied relating to ethical approvals (*i.e.*, approving body and any reference numbers):

Ethics committee approval is not needed for regular or scientific fishing in Iceland (The Icelandic law on Animal protection, Law 15/1994, last updated with Law 55/2013).

## Field Study Permissions

The following information was supplied relating to field study approvals (*i.e.*, approving body and any reference numbers):

All fishing in Lake Þingvallavatn was done with permissions obtained from the farmers in Mjóanes and from the Þingvellir National Park Commission.

Fishing in Hestvatn was done with the permit and help of Sigrún Reynisdóttir (landowner at Eyvík farm).

Sampling in other rivers was done within the Freshwater Institute science monitoring regime, and the Hengill volcano streams were surveyed with Gísli Már Gíslason (professor emeritus).

A scientific salmonid fieldwork permit was issued by the Directorate of Fisheries (#0460/2021-2.0 and #0042/2014-2.13).

## DNA Deposition

The following information was supplied regarding the deposition of DNA sequences:

The raw sequences are available at the NCBI Sequence Read Archive: PRJNA788782.

## Data Availability

The raw sequences are available at the NCBI Sequence Read Archive: PRJNA788782.

The code is available at GitHub and Zenodo:

- https://github.com/marclagu/ddRADseq_brown_trout_Iceland.git.
- https://doi.org/10.5281/zenodo.7637598.

dDocent scripts are available at: http://www.ddocent.com/downloads/ and https://github.com/jpuritz/dDocent (https://doi.org/10.7717/peerj.431).

## Supplemental Information

Supplemental information for this article can be found online at http://dx.doi.org/10.7717/peerj.15985#supplemental-information.

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
