# Peer review of "Genetic structure and relatedness of brown trout (Salmo trutta) populations in the drainage basin of the Ölfusá river, South-Western Iceland"

_PeerJ, doi:10.7717/peerj.15985_

## Round 0.1 · original submission · Minor Revisions

· Academic Editor

Minor Revisions

I have now received three reviewer reports for your paper. They are all constructive in nature and I feel addressing them will enhance the clarity and presentation of your paper.

Reviewer 1 ·

Basic reporting

The work by Lagunas et al describes population genetic analysis of brown trout at small geographical area in Iceland using 2,597 SNPs genotyped using ddRAD approach. It is generally well written and easy to follow, although in places it is quite long, the scope of this work is rather small and sometimes it is hard to infer the broader implications of this study. However, understanding genetic structuring in this systems will definitely help authorities in their efforts of conservation and management.

Fig. 1. The labels in this map should match the other figures in the manuscript (e.g. The authors use three letter code and specific colors in Fig. 2,3 and 6. Why not to use the same labelling and coloration to improve the clearness and readability.

Lines 625-616: there are several more recent papers by Anssi Vainikka group on genetic architecture of migratory tendency in brown trout which should be informative and relevant in this context.

Experimental design

Experimental design is sufficient for answering proposed question. Laboratory, bioinformatic and genetic methods are described in detail.

Lines 289-290: I was surprised that that authors used de novo approach for running Stacks despite the availability of Brown trout reference genome (Hansen et al. 2021; 10.12688/wellcomeopenres.16838.1) and benefits of using mapping to reference genome rather than de novo approach (Shafer et al. 2017; https://doi.org/10.1111/2041-210X.12700). The authors should at least justify why they used this approach.

I am curious if some of the juveniles used in this work could represent close relatives (given the small Ne estimates) and whether this may influence the diversity estimates..

Lines 374-382: The authors used heterozygosity excess approach for detection of recent bottlenecks. I curious why the authors chose to use step-wise mutation model (SMM) and two-phased model of mutation (TPM) which should be suitable for microsatellite markers but not for SNP-like data.

Line 223: Given that gill-net fishing procedures have been described very thoroughly, I was expecting the same for electro-fishing. Please provide at least the electrofishing equipment details.

Validity of the findings

Lines 520-521: I am curious why the effective population size estimates were so low, considering that the abundance of fish seems to have increased a lot over the recent years. If indeed the Ne estimates are so low, one expect that drift has a massive role shaping the genetic variation of studied trout populations and potentially overriding the effect of weak to moderate selection.

Lines 744-744: Perhaps it is useful here to acknowledge other suggested conservation strategies which are based on the evaluation of the relative roles of different evolutionary forces shaping the gene pools (Swatdipong et al. 2009, https://doi.org/10.1186/1742-9994-6-6) .

Additional comments

Abstract. Please consider removing "Using these data" in backgrounf of Abstract as you do not describe the data in previous sentences.

Lines 81-83: Please elaborate and/or provide references claiming that the costs of surveying genetic diversity in populations have decreased dramatically. I believe that the authors mean here that genome-wide screens have became feasible and cost per genotype has been decreased. However, in terms of screening tousands of individuals, one could even argue that per sample cost of creening allozymes is probably still lower than for ddRAD.

Line 108: Consider rephrasing: "but yield to Atlantic salmon"
Line 111: Consider rephrasing: "sported individuals"
Lines 144-146: Is there any archived scale/otolith records available in order to directly evaluate the effect of dam construction?

Reviewer 2 ·

Basic reporting

The introduction is interesting, but long and in need of some restructuring. In general, fewer words could be used to convey more or less the same information. Specifically, the details on lines 171-193 (in the reviewed pdf) should be shortened and/or moved to a supplement. I also suggest some basic information of high relevance to be introduced earlier. For example, the historic existence in L. Thingvallavatn of a downstream migrating population (spawning in R. Efra Sog) is mentioned first at line 147, although it would have been good to know when reading lines 114-124. Further, I suggest basic information on natural migration obstacles and the distribution of potential sea trout in the system (lines 159-171, 194-199) to be mentioned before describing potential effects of the man-made dams on the genetic population structure.

I also found the results section unnecessary long. It could be condensed rather much without the omission of any key information. For example, I do not see a need for describing results for K=6 and 7 at lines 454-463; concerning the results shown in figures 4 and 5, I would focus mainly on K=8 (perhaps with a few mentions of K=6-7). Lines 518-520 contain information already given in M&M that could be omitted. Further, some comments on specific results should be moved to the Discussion. It is a bit unusual to start with results on “Structure and gene flow” (present 3.1) before presenting “genetic diversity” (present 3.2). My suggestion is to simply switch the order of these two sub-sections, and begin with the basic diversity estimates that are based on sampling sites entirely (Table 3), and thereby can be presented before the mentioning of results on the genetic population structuring.

The Discussion is also long, but this is less of a problem (although I below suggest that a stricter focus on results for trout in the Ölfusá watershed could by a way to streamline and shorten the text somewhat). However, I didn’t see the need for all details about stocking on lines 717-727 - is all that information needed?

Experimental design

Although being a descriptive study, it has clear aims of high relevance for conservation and management. The sampling scheme fits well with the specific research questions and the distribution and history of brown trout in these neighboring water systems (for which no previous genetic studies exist).

The molecular methods used appear appropriate and up to date, although I have no personal experience of the ddRADSeq technique (thus, another reviewer with such expertise may have identified critical questions). Also, the statistical treatment seems accurate and motivated, but see comments below on the estimation of effective size and the interpretation of those estimates.

Validity of the findings

The estimation of effective population size and how those estimates are presented and discussed, raised some questions. First, depending on how many year classes are included in the analyzed sample, effective size estimates from the LD-method may reflect Ne (per generation), Nb (per year), or something in between. For a species like brown trout (with overlapping generations and repeat spawning) Nb is expected to be lower than Ne (Waples et al. 2013; Proc. Royal Soc. B 280:20131339). Further, if (when) sampling from a mixture of individuals from genetically distinct local populations, Ne-estimates derived with the LD-method may show a downward bias due to the additional LD expected in such mixtures (beyond that reflecting low effective size(s)). In the present case, it remained highly unclear how much of the variation in the “Ne-estimates” obtained may reflect true population differences or merely differences in the age structure (or degree of population mixture) among the samples analysed. At least a mentioning of these (indeed complex) aspects appears warranted in the Discussion.

It surprised me that the term “meta-population” was only mentioned once - in the very last sentence of the Discussion (line 754). For example, recent theoretical results (Ryman et al. 2019: Molecular Ecology 28:1904-1918) have shown that global Ne in a meta-population (consisting of several local populations connected by some gene flow) may largely prevent loss of gene diversity over time, despite low local Ne:s. A meta-population structure may explain some of the present results (e.g. levels of genetic diversity, no sign of a clear bottleneck, etc.) and should be mentioned earlier in the discussion (e.g. in connection to lines 670-682). Another aspect worth stressing further in a study like the present one is that possibilities for continued gene flow may be as important for genetic conservation as is the protection of local spawning populations (and their local Ne:s).

Is it likely that the local large-sized (?) population from L. Thingvallavatn that used to spawn downstream in R. Efra Sog has gone extinct, and if so, may this be a (main) reason for the declining catches in the past? From reading here and there, this sounds likely, but it is not mentioned or discussed. One place where this question could be brought up is, for example, in connection to the discussion on lines 572-577.

I wondered about the removal of 5% of all markers (those with the highest FSTs) mentioned on lines 364-365). Why this proportion and what may be the consequences if removing neutral loci that because of random genetic drift may display the highest levels of allele frequency differences? I missed a motivation for this decision, or preferably, results from an statistical outlier test(s).

Have the deviations between HO and HE mentioned at line 496 been tested statistically (no such results are mentioned in the text or shown in Table 3)? If not, why? To me, that would be standard procedure.

Details:

- Line 211: add the total number of individuals (“We sampled a total of xx trout from …”.
- Line 227: add “adults” (if correct).
- Line 237: add if the Scottish population is an anadromous or a resident one.
- Line 357: explain what is meant by “A non-parametric estimation of NE“.
- Line 370: clarify where (in what lake, catchment, etc.) catches declined in the 1980s.
- Line 386: add information that “mutation rate” here refers to SNP-markers (in trout).
- Line 389: clarify that “this analysis” is referring to the bottleneck test (if correct).
- Line 396: Add parenthesis with original total sample size: “… from 317 individuals (out of xxx originally analysed)”
- Line 438: clarify that “low pairwise FST values” here refer to comparisons among themselves (i.e. within the group BELOW).
- Line 538-539: add that the “seven clusters” mentioned (i.e. 7 out of 8 identified in the K=8 analysis) refer to clusters seen in Icelandic trout.
- Line 571: Clarify that “cold water spring sites” refer to spawning sites located “within the lake“.
- Lines 622-625: In addition to the suggested explanation, perhaps the level of gene flow among anadromous Icelandic populations is larger than downstream gene flow (if some such exists)?
- Lines 637-642: Here perhaps some of the older studies by Jonsson et al. of trout in the Norwegian “Vagnsvatnet system” could be cited, where local trout populations have been shown to display varying degrees of migratory behaviour (from residency to partial or complete lake and/or sea migration)?
- Line 698: It says “three fish from Innstidalur“, but according to Table 1 there are 6 fish from this location (HIN). Is the table showing the number of genes (rather than individuals)?
- Fig 1: It would assist reading if names were added to the map (and/or letter codes to the text; see below).
- Fig 3: The dendrogram was hard to read, despite color-coding. Partly this was because of poor graphic resolution on the pdf that I reviewed. Anyhow, I would anyhow suggest adding sample letters (or names) to the respective clusters, to facilitate reading.
- Table 1: Add the sum of individuals.
- Table 2: I missed statistical tests of these pairwise FSTs. If all were significant (which I could presume) it may be enough to mention this in the text. Otherwise P-values could be added in the table.

Additional comments

I found this study interesting and relevant, although as outlined above, the text is rather long and could benefit from some restructuring. One reason why the text partly feels a bit scattered is that the authors try to comment on many results for several different (nearby) waters. An alternative, to make the message shorter and more streamlined, could be to focus more strictly on trout in the Ölfusá watershed (i.e. L. Thingvallavatn with its inflows and outflow) and treat the other included populations (i.e. those from Hengladalsa, Hestvatn, and Leirvogsvatn) merely as comparative references.

In addition, parts of Results and Discussion were hard to read, as at many places only full names are referred to whereas abbreviations are (mainly) used in figures and tables (but see e.g. lines 403-405). Thus, I spent (too) much time trying to understand which river or lake is which one. I suggest consistent use of sample codes in the text, and when possible also full names (e.g. “Hestvatn (HES)…”).

Finally, at some places, the word "population(s)" is used in a way that makes the message unnecessarily vague (e.g. lines 443-444, 494-497, 499, 510-511). Although this term can be used in a broader non-genetic sense (e.g. "the trout population of Lake Thingvallavatn", line 12), one of the main purposes of this study is to identify (genetically homogeneous) local spawning populations, especially within and around L. Thingvallavatn. Therefore, I recommend the authors to be a little more careful about how and when this term is used, depending on the context.

Reviewer 3 ·

Basic reporting

This manuscript describes a study of genetic diversity, assessed by ddRADseq, among brown trout populations in Iceland. The study is fairly extensive, covering 4 lakes and 12 streams with a history of postglacial colonization, some of which also have been impacted by humans in the form of dam building and restocking attempts. The presentation is largely descriptive, relating observed genetic structures to hydrographic features. Tests for putative population bottlenecks and estimates of genetically effective populations sizes are also provided.

The manuscript is well written and the study appears to be competently carried out. The section on population structure (section 3.1) is fairly long (4 pages + figures and tables) and somewhat tedious in the amount of details. Similar with the Discussion (4.1: about 4 pages), which makes it a bit difficult to get a good overview. The authors may consider condensing these sections and omit details that are secondary to the main aims of the study. In contrast, I found Fig. 5 particularly helpful in visualizing the major genetic groups and their relation to hydrogeographic features in the study area. The authors may consider also putting heterozygosity (He) or nucleotide diversity on a map, instead of in the box plot as in the present Fig.6, to add geographic context. As waterfalls seem to be a major isolation factor it might be useful to have those added, as in Fig. 1.

The Discussion is weak on methodological evaluation and criticism. My impression is that admixture estimates are interpreted with perhaps too little concern for uncertainties and ambiguities in the results. Do "admixed" lakes/rivers harbor a mix of genetically different populations or just a single population that for historical reasons is somewhat intermediate in allele frequency? The distinction could be important both for management purposes but also for evaluating estimates of Ne (below).

Estimates of effective population size in this study make use of the so-called linkage disequilibrium (LD) method. This method, like others, builds on a number of assumptions that could introduce bias into the estimates if not met in practice. Given the extremely small Ne-estimates presented here (more than half the estimates were < 20: Table 4), downward biases seem likely. In particular, an Ne of only 3.4 as estimated for Leirvogsvatn seems unrealistically low as it implies an increase in inbreeding and loss of genetic variability of 1/(2*3.4) or 15% per generation, which is hardly sustainable. Reference to published estimates of Ne in brown trout or other salmonid populations could put these estimates in perspective. Potential sources for downward bias in estimated Ne could be physical linkage among loci and/or mixed samples from genetically divergent populations. Physical LD could be identified either by mapping loci to the brown trout reference genome (GenBank accession number GCA_901001165.1) and identifying pairs that map closely, say within a few kilobases, or by checking for locus-pairs that consistently display elevated LD in multiple populations. Admixture LD could perhaps be checked for by comparing estimated Ne to the estimated admixture rates among populations.

Finally, have potential effects of data filtering (section 2.4.2) on the results been considered? For example, could filtering of loci that deviate from Hardy-Weinberg genotype proportions (l.321) risk hiding a real signal of population admixture (Wahlund effect)? Also, I'm not entirely clear on whether data (haplotypes?) for the bottleneck test (section 2.4.5) were filtered on minor allele frequency and, if so, what would be the impact on this test?

The following are minor comments:

l.136: How come an increase of juveniles in River Öxará signifies that the stocking in Þingvallavatn and Ölfusvatnsá was successful?

l.389-391: So, the bottleneck test was carried out on haploid data and not on di-allelic SNPs, correct? What about minor allele frequency filtering?

l.496: If deviations from HWE are significant this should be pointed out. It is not clear from the small samples (cf. Table 3) that deviations are large enough to be statistically significant.

Figure 1: It is awkward do look up the the many letters and digits on the map: Would it be possible to put (abbreviated) names on the map instead?

Experimental design

No comment

Validity of the findings

No comment

---

## Round 0.2 · Minor Revisions

· Academic Editor

Minor Revisions

The clarity of the work has improved following revision, but so has the length of the manuscript. Consider reducing the verbosity and finer details not directly related to the hypothesis that you test. Also consider the points that the reviewer raises regarding the effective population size and the number of breeding individuals per year, and the earlier literature pertaining to this.

Reviewer 3 ·

Basic reporting

This manuscript describes population genetic structure within a drainage system on Iceland consisting of multiple stream and lake brown trout populations. This revision addresses most of the concerns raised by the reviewers of the original submission, although suggestions for exploring potential biases in estimates of Ne were not heeded (see below).

All reviewers also suggested cutting down on the amount of detailed descriptions but the attempts to do so did not actually reduce manuscript length (on the contrary, the Introduction increased by nearly 1 page and the Discussion by 2.5). Anyhow, the presentation is easier to follow now, especially with the updates on the figures.

The following are minor points:

l.387-388: A minor allele frequency of 1% can only be achieved for samples larger than 50, so using this limit for most of the present samples (cf. Table 1) would imply no MAF filtering at all. Was that the intention?

l.750: I doubt the generality of this statement: "Long isolated populations are prone to have low effective population sizes."

Figures 2 and 3: Axis numbers and legends are too small to read.

Figure 5: Explain asterisks on He/Ho numbers.

Experimental design

No comment

Validity of the findings

Recalling that the LD method assumes physically unlinked loci, I wonder what is meant by the sentence "multiple SNPs per RAD locus were included in the analysis" (l.389)? Clearly, if the analysis is done at the SNP-level known physical linkage should be removed before Ne is estimated from SNP genotypes. No attempts seems to have been made in this revision to check if the low Ne-estimates could be biased due to physical linkage or to population admixture in the samples. Given the significance given to the low Ne estimates, as judged by two full pages of discussing biological causes and consequences (l.725-773), I find this omission a serious weakness. The authors have added some general comments regarding Ne and Nb (l.756-758), and cite a study on a different species (brook trout) apparently in support of very low Ne-values. However, Ne (and Nb) in brown trout is known in considerable details from previous studies that are currently not cited (try searching for brown trout effective population size in Google scholar), and few of the estimates from this literature overlap the several very small values reported herein. Hence, I suspect there may be an issue with downward bias in the present estimates.

---

## Round 0.3 · accepted · Accept

· Academic Editor

Accept

The authors have addressed the points raised by the reviewer to an appreciable degree, and I am happy to accept this paper in its current form.